



**Analyzing ozone variations and uncertainties at high latitudes during Sudden**
**Stratospheric Warming events using MERRA-2**
**Shima Bahramvash Shams**[1,2], Von P. Walden[1], James W Hannigan[2], William J. Randel[2],
Irina V. Petropavlovskikh[3,4], Amy H. Butler[5], Alvaro de la Cámara[6]
[1] Washington State University, Pullman, WA, United States,
[2] NCAR, National Center for Atmospheric Research, Boulder, CO, United States,
[3] Cooperative Institute for Research in Environmental Sciences, University of Colorado, Boulder, CO, USA
[4] National Oceanic and Atmospheric Administration, Global Monitoring Division, Boulder, CO, USA
[5] National Oceanic and Atmospheric Administration, Chemical Sciences Laboratory, Boulder, CO, USA
[6] Dept. Física de la Tierra y Astrofísica, Universidad Complutense de Madrid, Madrid, Spain
Corresponding author: Shima Bahramvash Shams, s.bahramvashshams@wsu.edu
**Abstract:**
Stratospheric circulation is a critical part of the Arctic ozone cycle. Sudden stratospheric
warming events (SSWs) manifest the strongest alteration of stratospheric dynamics. Changes in
planetary wave propagation vigorously influence zonal mean zonal wind, temperature, and tracer
concentrations in the stratosphere over the high latitudes. In this study, we examine six major
SSWs from 2004 to 2020 using the Modern-Era Retrospective analysis for Research and
Applications, Version 2 (MERRA-2). Using the unique density of observations around the
Greenland sector at high latitudes, we perform comprehensive comparisons of high latitude
observations with the MERRA-2 ozone dataset during the six major SSWs. Our results show that
MERRA-2 captures the high variability of mid stratospheric ozone fluctuations during SSWs over
high latitudes. However, larger uncertainties are observed in the lower stratosphere and
troposphere. The zonally averaged stratospheric ozone shows a dramatic increase of 9-29% in total
column ozone (TCO) near the time of each SSW, which lasts up to two months. The SSWs exhibit
a more significant impact on ozone over high northern latitudes when the polar vortex is mostly
elongated as seen in 2009 and 2018 compared to the events in which the polar vortex is displaced





towards Europe. The regional impact of SSWs over Greenland has a similar structure as the zonal average, however, exhibits more intense ozone anomalies which is reflected by 15-37% increase in TCO. The influence of SSW on mid stratospheric ozone levels persists longer than their impact on temperature. This paper is focused on the increased (suppressed) wave activity before (after) the SSWs and their impact on ozone variability at high latitudes. This includes an investigation of the different terms of tracer continuity using MERRA-2 parameters, which emphasizes the key role of vertical advection on mid-stratospheric ozone during the SSWs.

## 1. Introduction

Stratospheric ozone can modulate the radiative forcing of climate and Earth's surface temperature (Haigh, 1994; Ramaswamy et al., 1996; Smith and Polvani, 2014; Calvo et al., 2015; Kidston et al., 2015; Nowack et al., 2015; Romanowsky et al., 2019). High latitude stratospheric ozone influences tropospheric climate, surface temperature of lower latitudes, El Niño-Southern Oscillation (ENSO) events, and the North Pacific Oscillation (NPO) (Baldwin and Dunkerton, 2001; Ineson and Scaife, 2008; Cagnazzo and Manzini, 2009; Karpechko et al., 2014; Xie et al., 2016). Thus, it is important to have a thorough understanding of high latitude ozone variations.

Dynamical variability plays a critical role in fluctuations of stratospheric ozone (Holton et al., 1995; Fusco and Salby, 1999; Rao et al., 2004; Bahramvash-Shams et al., 2019). Planetary waves modulate poleward ozone transport through the Brewer-Dobson circulation (BDC) (Lindzen and Holton, 1968; Holton and Lindzen, 1972; Wallace, 1973; Holton et al., 1995). The BDC transports tropical stratospheric air parcels with high ozone concentrations to the high-latitude stratosphere, causing ozone accumulation during winter and peak values in the spring (Rao, 2003; Rao et al., 2004). Sudden stratospheric warming events (SSWs) are the largest alterations of stratospheric circulation during wintertime and a significant factor in the interannual variability of stratospheric transport (Schoeberl, 1978; Butler et al., 2015; de la Cámara et al., 2018a).



SSWs are defined as an abrupt and intense stratospheric temperature increase that coincides
with a reversal of the climatological westerly wind circulation (Scherhag, 1952, Baldwin et al.
2021).  Although the current understanding of the mechanisms that induce SSWs is still uncertain
(de la Cámara et al., 2019; Lawrence and Manney, 2020), increased vertical propagation of
planetary-scale waves from the extratropical troposphere into the stratosphere over high latitudes
is closely related to these abrupt events (Matsuno, 1971; Schoeberl, 1978; Scott and Polvani,
2004). However, the occurrence of SSWs is shown to be sensitive to many other factors such as
lower stratosphere conditions, the geometry of the polar vortex, the gradient of potential vorticity
(PV) at the edge of the polar vortex, and synoptic systems at lower altitudes (Tripathi et al. 2015,
de la Cámara et al., 2019; Lawrence and Manney, 2020). Changes in momentum deposition
associated with these processes leads to the rapid deceleration and disruption of the stratospheric
polar vortex, typically by either splitting the vortex into two smaller lobes or displacing the vortex
off the pole (Matsuno, 1971; Polvani and Waugh, 2004; Charlton and Polvani, 2007). The altered
circulation during SSWs impacts the transportation of trace gases (Randel 1993, de la Cámara et
al., 2018b), can influence tropospheric weather and climate (Baldwin and Dunkerton, 2001; Butler
et al., 2017; Charlton-Perez et al., 2018, Butler and Domeisen 2021), and gravity waves over the
Arctic (Thurairajah et al., 2010) and consequently the pole-to-pole circulation (Houghton, 1978;
Fritts and Alexander, 2003). SSWs are one of the strongest manifestations of atmospheric
coupling. These large-scale altered circulations perturb the mesosphere by cooling it and
consequently lowering the stratopause by up to 30 km (Manney et al., 2008b). Dynamical coupling
between the stratosphere and troposphere is another important consequence of SSWs with
implications for surface climate predictability on subseasonal timescales (Baldwin and Dunkerton
2001, Butler et al. 2019).
The Modern Era Retrospective Analysis for Research and Application, version 2
(MERRA-2) is used to investigate ozone fluctuations during SSWs. Previous validation of
MERRA-2 ozone data with ozonesondes and satellite data over the South Pole and midlatitudes
has shown good correlation (Gelaro et al., 2017; Wargan et al., 2017). However, MERRA-2 ozone
data are expected to have higher uncertainties over the northern high latitudes because of higher



dynamic variability in this region (Wargan et al., 2017). During SSWs, the alteration of dynamical processes causes dramatic variability in trace gas concentrations in the middle atmosphere. The complexity of altered dynamics of SSWs might introduce extra uncertainties into the numerical and assimilation models. The performance of MERRA-2 ozone products during SSWs has not been investigated in previous studies. It is essential to understand the performance of MERRA-2 ozone simulations during these anomalous events before using them for further analysis of ozone variations.

Various studies have focused on individual SSWs, their evolution, and dynamical characteristics (Siskind et al., 2007; Manney et al., 2008a; Coy et al., 2009; Manney et al., 2009b). Some studies have investigated ozone transport during SSWs based on modeling and simulation (Tao et al, 2015; de la Cámara et al., 2018 a and b; Oehrlein et al, 2020). A few studies used observations to investigate local (Flury et al., 2009; Scheiben et al 2012; Schranz et al 2020) or zonal impacts of specific SSW events on ozone (Manney et al 2009a; Manney et al, 2015). Numerical models have shown the influence of SSWs on stratospheric transport and mixing over high latitudes (Tao et al., 2015; de la Cámara et al., 2018b). This study investigates zonal average ozone variations (between 60 and 80 N) during six SSWs using the MERRA-2 dataset.

This study focuses on using observations and an assimilation model to analyze and compare the impact of SSWs on ozone from 2004 to 2020. During SSWs, MERRA-2 ozone data are compared with in situ and ground-based remote sensing observations from high northern latitudes. The advantage of an existing dense network of observations around the Greenland sector at high latitudes (Figure 1) provides an opportunity to explore the uncertainties of MERRA-2 ozone profiles over high latitudes during SSWs. These comparisons provide a thorough understanding of the uncertainties in the MERRA-2 dataset in this region and, in particular, during extreme dynamic events.

In section 2, MERRA-2 and other independent observations are described. The methodology of comparisons and dynamical analysis are presented in section 3. The results of the comparison between MERRA-2 and independent observation are discussed in section 4. The





evolution of each SSW and its impact on ozone are discussed in section 5. Discussion of transport
mechanisms is provided in section 6. Section 7 presents the conclusions of this research study.
**2. Data**

4         The Modern-Era Retrospective Analysis for Research and Application, version 2

(MERRA-2) from NASA's Global Monitoring and Assimilation Office (GMAO) uses the
Goddard Earth Observing System, Version 5 (GEOS5) atmospheric data assimilation system
(Molod et al., 2015; Gelaro et al., 2017). A variety of data sets and models are incorporated in
MERRA-2 to create 3-dimensional ozone datasets with a time-frequency of 3 hours, including
ozone and meteorological observations, global circulation models, and extended reanalysis
(Wargan et al., 2017; Gelaro et al., 2017). Total column ozone from the Solar Backscatter
Ultraviolet Radiometer (SBUV) (1980 to 2004) and the Ozone Monitoring Instrument (OMI)
(since 2004) and retrieved ozone profiles from SBUV (1980 to 2004) and the Microwave Limb
Sounder (MLS) (since 2004) are used to estimate ozone in MERRA-2 (Gelaro et al., 2017).
MERRA-2 data are available online through the NASA Goddard Earth Sciences Data Information
Services Center (GES DISC; http://disc.sci.gsfc.nasa.gov/ daac-bin/DataHoldings.pl).

16         MERRA-2 has been used extensively to study ozone trends and processes (Coy et al., 2016;

Knowland et al., 2017; Wargan et al., 2018; Shangguan et al., 2019). In this study, the ozone
dataset from the MERRA-2 reanalyses at a spatial resolution of $0.5° \times 0.625°$ will be used. To
investigate dynamical mechanisms, temperature, the northward wind (v), vertical pressure velocity
($\omega$), potential temperature ($\theta$, calculated from temperature and pressure), and potential vorticity
(PV) are extracted from the pressure-level MERRA-2 dataset.

22         In assimilated/reanalysis models such as MERRA-2, variations in models, methods of

analysis, and observations cause uncertainties in the products (Rienecker et al., 2011). Previously
MERRA-2 ozone data was validated using ozonesondes and satellite data from 2005 to 2012
(Gelaro et al., 2017; Wargan et al., 2017). MERRA-2 agreement with independent observations
has been improved since 2005 by assimilating OMI and MLS. Comparison with independent
satellite observations show an average standard deviation of the differences of 5% and 11% in the





upper and lower stratosphere, respectively (Wargan et al., 2017). The average standard deviation
of 20% has been reported for the comparison between MERRA-2 lower stratospheric ozone and
ozonesondes (Wargan et al., 2017). However, uncertainties are expected to be magnified at high
latitudes because of higher dynamical variability (Wargan et al., 2017). Moreover, the anomalous
atmospheric dynamics, displace/split polar vortex, and hemispherically asymmetric conditions
during SSWs may cause unusual nonlinearity in ozone flux/transport terms. Thus, it is important
to investigate the quality of MERRA-2 ozone simulations during highly altered circulations such
as SSWs. This study provides a comprehensive comparison using ground-based remote sensing
and in situ observations to MERRA-2 ozone datasets over northern high latitudes during SSWs.
We also use a uniquely dense network of observations in the high latitudes to study a region of the
Arctic that is climatologically important in terms of stratospheric circulation (Figure 1).

12        Ozonesondes have been used to monitor ozone for decades as the most direct measurement

of the vertical ozone profile (Tiao et al., 1986; Logan, 1994; Logan et al., 1999; Stolarski, 2001;
Gaudel et al., 2015; Bahramvash-Shams et al., 2019). Ozonesonde profiles provide a good standard
for validation because they have high accuracy, fine vertical resolution of less than 100 m, year-
round launches, and low sensitivity to clouds (McDonald et al., 1999; Ancellet et al., 2016; Sterling
et al., 2017).

18        In this study, ozonesonde measurements at Eureka, Ny-Ålesund, Thule, and Summit will

be used to investigate the uncertainties of MERRA-2. The locations of each station and the length
of the ozonesonde measurements at each site are shown in Figure 1 and Table 1. Most of the
ozonesonde measurements can be found at the World Ozone and Ultraviolet Radiation Data Centre
(WOUDC), while ozonesonde data in the United States is obtained from NOAA's Earth System
Research Laboratory including data from Summit Station, Greenland. The detailed description and
uncertainty estimation of ozonesonde measurements have been discussed in previous studies
(Komhyr, 1986; Johnson et al., 2002; Smit et al., 2007; Tarasick et al., 2016; Sterling et al., 2017).

26        In addition to ozonesondes, ground-based remote sensing data are also used in this paper

to study the uncertainties in the MERRA-2 dataset.  Retrieved ozone from ground-based Fourier





transform infrared (FTIR) interferometers have been used for long term ozone analysis (Vigouroux
et al., 2008; García et al., 2012; Vigouroux et al., 2015). In this study, ozone profiles retrieved
from FTIR at five high-latitude sites (Eureka, Ny-Ålesund, Thule, Harestua, and Kiruna) were
obtained from NDACC (Network for the Detection of Atmospheric Composition Change) and
used to validate MERRA-2. The location of each site is shown in Figure 1 and Table 1. These
datasets are available at http://www.ndacc.org.

7        The NDACC FTIR instruments measure solar radiation in a wide spectral bandwidth of

600-4500 cm$^{-1}$ at a high spectral resolution of 0.0035 cm$^{-1}$. The retrieval of ozone profiles from
NDACC FTIR instruments uses the optimal estimation method (Rodgers, 2000). NDACC
retrievals use the spectroscopic database from HITRAN 2008 (Rothman et al., 2009). To retrieve
trace gas information from the measured spectra using optimal estimation, additional information
is required to constrain the result and find the optimal answer. Meteorological parameters from the
National Centers for Environmental Prediction (NCEP) and monthly trace gas profiles from the
Whole Atmosphere Community Climate Model WACCM4 (Marsh et al., 2013) are used as a priori
conditions. More details of the NDACC ozone retrieval steps, configuration, and instrument
specifications are discussed by Vigouroux et al (2008; 2015). Having the ability to resolve the fine
structure of solar radiation spectra allows the retrieval of a variety of trace gases using the NDACC
solar FTIR. However, these instruments require sunlight and clear-sky conditions, which restricts
observations to the polar day at high latitudes.

20       The retrieved total ozone column and the stratospheric partial columns from FTIR are

expected to have uncertainties of 2% and 6%, respectively (Vigouroux et al., 2015; Bognar et al.,
2019). This study updates the uncertainties found by previous studies by adding additional years
of data and by focusing on three high latitude sites that contain both ozonesondes and FTIR
measurements. The FTIR ozone retrievals showed a high correlation ($\sim$ 90%) in comparison to
ozonesonde profiles measured at Eureka, Ny-Ålesund, and Thule, with uncertainties shown in
Table 1. Overall, the uncertainties are slightly higher than the averaged uncertainties reported by
Vigouroux et al (2015). This is more pronounced at Eureka due to the high solar zenith angle, and
the possibility that, at times, the FTIR views a slant path through the atmosphere that extends



through the edge of the polar vortex. More details of the ozone retrievals at Eureka can be found
in Bognar et al (2019). As shown in Table 1, the NDACC retrievals are biased high when compared
to the ozonesondes. Also, the bias is higher at Eureka (7%) than at either Ny-Ålesund (1%) and
Thule (3%). These biases and standard deviations (shown in Table 1) are less than the differences
between MERRA-2 and the ozonesondes (20%) discussed above, indicating that the NDACC
FTIR ozone retrievals can be used to increase the robustness of the uncertainty analysis of the
MERRA-2 ozone dataset.
**3. Methods**
In this section, the details of the different methods used in this study are discussed,
including the comparison methodology, detection of SSWs, and the derivation of dynamical
parameters used to investigate ozone transport.
To have comparable points, NDACC and in situ site locations, shown in Figure 1 and Table
1, are extracted from the nearest $0.5° \times 0.625°$ grid MERRA-2 ozone dataset. The nearest
instantaneous 3-hourly MERRA-2 ozone dataset is compared to the associated ozonesonde profile
and the FTIR-retrieved ozone. The MERRA-2 ozone data are compared to ozonesondes at the
model levels, up to the maximum measured altitude. However, the vertical resolution of the remote
sensing retrieval is often not similar to the model grid points, to apply the retrieval sensitivity to
the model level prior to the comparison, the averaging kernel is used through the smoothing
method (Rodgers and Connor, 2003). Averaging kernels characterize the vertical resolution and
sensitivity of FTIR instruments to the atmospheric ozone variability at various altitudes (Rodgers,
2000). Equation 1 shows how the averaging kernel is applied with the model data to account for
the sensitivity of retrievals (Rodgers and Connor, 2003), producing a smoothed ozone profile.
$$x_s = x_a + A\,(x_h - x_a) \qquad (1)$$
where $x_s$ is the final smoothed ozone profile, $x_h$ is the model estimated profile, and $x_a$ and $A$ are
the a priori and averaging kernel of the retrieval respectively. The smoothing method effectively
linearizes the ozone from the model using the averaging kernel of the retrieval around the a priori





information (Rodgers and Connor, 2003). MERRA-2 data are interpolated to the vertical grid of
the retrievals before Equation 1 is applied.

3       The high spectral resolution of the solar FTIR measurements makes it possible to retrieve

partial ozone columns in addition to the total column ozone. Based on the mean average kernels
at all 5 stations, four partial column ozone (PCO) are determined in this study over the following
altitude regions: ground-8 km, 8-15 km, 15-22 km, 22-34 km. The PCO amounts are also used to
analyze uncertainties in the MERRA-2 ozone dataset. The comparison results are discussed in
section 4.

9       There are a variety of definitions for detecting major SSWs (Charlton & Polvani, 2007;

Butler et al., 2015; Palmeiro et al. 2015). This study uses wintertime reversals of the daily-mean,
zonal-mean zonal winds at 60N and 10 hPa from the MERRA-2 dataset (Butler et al., 2017). The
dates of major SSWs since 2004 are calculated using MERRA-2 data following the method
described by Charlton & Polvani (2007) and are shown in Table 2. Table 2 also includes the
duration, magnitude of the easterly zonal wind, and the duration of polar vortex recovery for each
SSW; all information is derived from MERRA-2 data. It should be noted that the duration of the
easterly wind shown in Table 2 is not necessarily consecutive.
The dynamical transport mechanisms are also investigated for each of the major SSWs.
The zonal mean tracer concentration is a balance between transport processes and the chemical
sources and sinks as shown in the continuity equation of the Transformed Eulerian Mean (TEM)
(Andrews et al, 1987):
$$\bar{x}_t = -\bar{v}^* \, \bar{x}_y - \bar{w}^* \, \bar{x}_z + e^{Z/H} \Delta. M \; + P - L \; (2)$$
where $\bar{x}_t$ is the tracer tendency (in this case, ozone mixing ratio), $(\bar{v}^*, \bar{w}^*)$ are horizontal
and vertical components of the residual circulation, z = -H ln(p/p0) in log-pressure height using a
scale height H of 7 km, M is eddy transport vector, and P and L are chemical production and loss.
The overbars stand for the zonal average. Subscript symbols denote partial derivatives [with
respect to time (t) and height (z)]. The first two terms on the right-hand side of equation (2)



represent the contribution of advective transport on ozone changes. The vertical component of
residual circulation is the dominant contributor of advection ($\overline{w}^*$) and can be estimated using TEM
(Andrews et al, 1987):

$$\overline{w}^*=\overline{w} + \frac{1}{acos\phi(\varphi)}\partial_\varphi(cos(\varphi)\frac{\overline{v'\Theta'}}{\Theta_z}) \quad (3)$$

5       where v and w are the meridional and vertical winds, $\Theta$ is potential temperature, a is the

earth radius, $\varphi$ is the latitude. The prime denotes the departure from the zonal mean. The third
term on the right side of equation (2) shows the impact of eddy mixing on ozone transport. M can
be decomposed into vertical and meridional components $M_z$ and $M_y$ respectively: (Andrews et al.,

9   1987):

$$M_y= -e^{(-z/H)}(\overline{v'x'} - \frac{\overline{v'\Theta'}}{\Theta_z}\bar{x}_z) \quad (2.4)$$

$$M_z= -e^{(-z/H)}(\overline{w'x'} + \frac{\overline{v'\Theta'}}{\Theta_z}\bar{x}_y) \quad (2.5)$$

12       This study investigates the impact of the dynamics on polar ozone concentrations during

individual SSWs using MERRA-2 data. Because the impact of the chemical components on the
evolution of ozone during SSWs is a less important factor below 30 km (de la Cámara et al 2018b),
the dynamical analysis in this study will focus on altitudes below 30 km. Thus, the lack of net
chemical production in the assimilation model should not dramatically impact our conclusions.
**4. Comparison of Observations with MERRA-2**

18       In this section, the results of the comparisons between MERRA-2 and observations from

ozonesondes and FTIR retrievals during SSWs are discussed. Ground-based observations provide
an excellent baseline to assess climate models and assimilated models. However, the use of
ground-based observations to directly study the impact of SSWs is challenging because of the
sparse site locations, coarse time resolution (ozonesondes), and limited clear-sky conditions and
sunlight (FTIR). In this study, we take advantage of a dense network of observations to assess the
performance of MERRA-2. The use of MERRA-2 allows us to investigate the fluctuations over





1 the entire Arctic with consistent temporal and spatial resolution. To visualize the observation

2 frequency and the overall performance of MERRA-2, the time series of PCO from MERRA-2 3-

3 hourly data and ozonesondes and FTIR from winter 2007 to spring 2009 are shown in Figure 2.

4 Two major SSWs occurred during this time period. To exhibit a consistent time series and to avoid

5 the impact of the variability of maximum height of the ozonesondes, PCO from the ground to 20

6 km is shown. Figure 2 shows the high temporal frequency of the FTIR retrievals compared to

7 ozonesondes during polar day, the consistent frequency of ozonesondes throughout the year, and

8 the gap in solar FTIR retrievals at high latitudes during polar night. The results indicate a good

9 overall agreement of MERRA-2 with observations. The sparsity of FTIR ozone retrievals at Thule

10 in 2008 was due to instrument issues. To have a more clear understanding of the uncertainties in

11 MERRA-2 estimations, more quantitative comparisons are needed.

12  To investigate the uncertainties of MERRA-2 ozone data during the highly anomalous

13 conditions during SSWs and to consider the enduring impact of SSWs on trace gases, comparisons

14 are performed from 1 December to 1 May for all six events. The results of comparisons between

15 ozonesondes and MERRA-2 are depicted in Figure 3. The mixing ratios from the ozonesonde

16 profiles are subtracted from the MERRA-2 profiles and divided by the ozonesonde concentration

17 at the corresponding levels and are shown as the difference ratios. The mean and standard deviation

18 of the difference ratios in three layers PCO [ground to 5km (G-5km), 5km-10km, 10km-30km],

19 are also reported in Figure 3. The PCO difference ratio is estimated as PCO from MERRA-2 minus

20 ozonesonde PCO divided by ozonesonde PCO for each of the 3 layers. These layers indicate

21 different performance of MERRA-2 by height and show the effect of atmospheric pressure on the

22 contribution of each level to the total ozone column. The G-5km layer includes the troposphere,

23 the 5km-10km includes upper troposphere-lower stratosphere (UTLS), while the 10-30km layer

24 includes the middle and lower stratosphere. The partial column is calculated only up to the altitude

25 of the balloon burst of the ozonesonde, if the burst height is below 30 km.

26  Large difference ratios between MERRA-2 and the ozonesondes near the surface indicate

27 a well-defined high bias in MERRA-2 at Ny-Alesund and Eureka. The occasional extreme low

28 ozone mixing ratios observed in the lower atmosphere and near the surface are linked to catalytic



reactions involving bromine. This chemical ozone depletion is more common at Arctic sites near
the ocean (Tarasick and Bottenheim, 2002). MERRA-2 is shown to be unable to retrieve the
extreme low ozone values near the surface.

4         Overall, the variability of the difference ratios at lower altitudes are larger (Figure 3). Ny-

Alesund and Eureka show 5%(±23%) and 18%(±26%) mean (±std) difference ratio at G-5km.
However, the G-5km layer, on average, contains less than 20 DU, which is less than 6% of total
column ozone (TCO). PCO of the G-5 km layer is only 1.5% of TCO at Summit Station where the
site elevation is 3.2 km. The PCO difference ratio at Summit station shows very small bias with a
standard deviation of ±15%.

10        The positive bias decreases higher in the troposphere, and the scatter plot shows negative

difference ratios. From 5 km to 10 km, a non-significant (higher standard deviation) negative mean
bias exists at all sites. The mean PCO difference ratios from 5 km to 10km are -8%(±13%), -
15%(±15%), and -8%(±16%) at Summit Station, Ny-Alesund, and Eureka.

14        The MERRA-2 ozone data between 10 and 30 km are highly correlated with the

ozonesondes with $R^2$ > 90% (not shown). From 10 to 15 km, the difference ratios are slightly
positive and, above 15 km, a negligible bias and low standard deviations are observed. The mean
PCO difference ratio in the 10-30 km layer is equal to or less than 3% (±7%) at all stations. The
differences between 10 and 30 km are more impactful in TCO uncertainty analysis because this
region contains most of the column ozone. (The average PCO for each layer is reported in Figure

20   3.)

21        Figure 4 summarizes the comparison between the MERRA-2 and the FTIR retrievals for

December 1st to May 1st for all six SSW years. The partial column comparisons for ground to 8
km, 8-15 km, 15-22 km, and 22-34 km are shown. Here the partial columns are defined based on
the averaging kernel of the NDACC retrievals. The mean and standard deviation of difference
ratios, the mean PCO for each layer are shown in Figure 4.





The layers between 15-22 km and 22-34 km contain the most column ozone with averages
of 146 DU and 101 DU, respectively. MERRA-2 and the FTIR retrievals have good agreement in
these layers with difference ratios of -2%±5% and -4%±5%, respectively.
In the lowest layer, the differences are the largest with a standard deviation ratio of higher
than 15% at all stations and mean differences in the range of -7% to 3%. Large differences are
observed between 8-15 km, where MERRA-2 estimates 7%-13% more ozone than the FTIR
retrievals, and the standard deviations are large. Large differences and standard deviations below
15 km indicate that higher uncertainties exist in both the FTIR retrievals and the MERRA-2
estimation.
In conclusion, when compared to observations, MERRA-2 captures large fluctuations in
middle stratospheric ozone at high northern latitudes during winters and early spring that are
impacted by SSWs. However, the differences in the lower stratospheric and tropospheric layers
exhibit larger values. The higher uncertainties below 15 km during the five months impacted by
SSWs are consistent with higher uncertainties in MERRA-2 in these layers year-round, as seen in
previous studies (Gelaro et al., 2017; Wargan et al., 2017). The agreement between MERRA-2
ozone with observations during SSWs motivates the use of MERRA-2 dataset to further
understand the mid-stratospheric ozone fluctuations during SSWs. The maximum height of
ozonesondes is around 30-35 km and ground-based remote sensing loses sensitivity with
increasing altitude, thus this study cannot improve previous research on upper stratosphere where
higher uncertainties were reported compared to the mid stratosphere. Because more than 75% of
ozone molecular density exists in the middle stratosphere (15 to 30 km), the total column
uncertainty is dominated by uncertainties in mid-stratospheric layers. In the following section, we
discuss ozone variability in the total column and the vertical profile up to 60 km, while our primary
analysis is focused on the mid-stratospheric layers because this is where the measurements are
most reliable which also has the dominant density of ozone.



## 5. SSWs and their impact on ozone

Disturbances in stratospheric circulation have an impact on stratospheric trace gas concentrations. Consequently, the temporal changes of trace gas concentrations can provide a better understanding of atmospheric circulation including vertical and horizontal transport (Manney et al., 2009a). In this section, the impact of altered circulation patterns on ozone is analyzed, and by investigating the evolution of polar vortex and temperature more detailed characterization of ozone variability is provided.

To understand the rapid alteration of ozone and the average position of the polar vortex before and after each SSW, the anomaly of total column ozone (TCO) and the average Ertel's potential vorticity (PV) contours of 60 and 80 (105 K m2 Kg-1 s-1) at isentropic level with the potential temperature of 850 K for 15 days preceding and 15 days after each of SSWs are shown in Figure 5. In the following parts the main characterization of each SSWs, the evolution of polar vortex, and TCO changes are discussed.

2006: On 21 January 2006, the second strongest and prolonged major SSW since 2004 was detected (Table 2, Siskind et al., 2007; Manney et al., 2008b; 2009a). The easterly zonal mean zonal wind lasted 26 days. Prior to the major SSW, a minor SSW was detected on 9 January (Manney:2008b, Manney:2009a). The polar vortex moved toward Siberia and receded away from Greenland during the minor warming. The polar vortex then displaced westward and equatorward toward northwestern Europe before the major SSW as shown in Figure 5a.

2008: The dynamical circulation was quite variable during winter 2008. Two minor SSWs in mid and late January and one major SSW in late-February are recorded in 2008 (Goncharenko and Zhang, 2008; Flury et al., 2009; Thurairajah et al., 2010; Korenkov et al., 2012). The easterly winds lasted 15 days during the major warming on 22 February. This event is recorded as the latest in the winter season and the least prolonged among the six SSWs considered in this study (Table 2). The polar vortex is displaced mostly over northwest Europe during the development of the SSW in 2008 as shown in Figure 5c. The polar vortex displacement over Europe led to ozone


depletion and the enhancement of stratospheric water vapor over northern Europe by mid-February
(Flury et al, 2009).

3       2009: Following an undisturbed and cold early winter, the strongest and most persistent

SSW among this study events occurred on 2 January 2009 as shown in Table 2 ( Manney et al.,
2009b; Harada et al., 2010; Lee and Butler, 2019). The extended elongated shape of the polar
vortex before the SSW can be seen in Figure 5e, which resulted in a split vortex. The prolonged
SSW in late January recorded 30 days of easterlies at 10 hPa with a maximum magnitude of 29
m/s (Table 2).

9       2013: The atmospheric disruption associated with the major SSW on 6 January 2013

displaced the polar vortex toward Europe (Figure 5g) and eventually caused the stratospheric polar
vortex to split and smaller vortices to appear over Canada and Siberia in mid to late January
(Manney et al., 2015). The isolated, offspring vortex lasted for more than two weeks over Canada
as shown in Figure 5h.
2018: A major SSW was detected on 12 February 2018. However, the disturbed circulation
started in January, with 8 days of zonal wind deceleration occurring in mid-January (Rao et al.,
2018). The elongated pattern of PV contours from Europe to eastern Canada shown in Figure 5i
indicates a highly disturbed vortex prior to the major SSW and finally led to the vortex split
(Karpechko et al., 2018; Rao et al., 2018; Butler et al. 2020). The split vortices were located over
Canada/northwest US and northwestern Europe and lasted for almost a week after the detected
SSW. The signal of the offspring vortex after the SSWs over Canada is visible in Figure 5j. The
major SSW and its associated polar vortex dispersal caused record-breaking cold surface
temperatures in northwest Europe (Greening and Hodgson, 2019).

23       2019: The major SSW on 2 January 2019 (Butler et al. 2020; Rao et al., 2019, Schranz et

al., 2020) is the earliest in the winter season and weakest in the magnitude of reversal among the
most recent six events studied here (Table 2). The polar vortex was displaced toward Europe before
the major SSW occurred (Figure 5k). The continuous wave activity caused a vortex displacement





to be followed by a split vortex. The resulting vortices were located over the northeastern US and
northwestern Europe as shown in Figure 5l.

3        As shown in Figure 5, the vortex displacement toward the southeast (Europe) prior to the

major SSW as seen in 2006, 2008, 2013, and 2019 (hereafter the displaced vortex SSWs) caused
an early positive ozone anomaly in the region outside of the vortex which includes parts or all of
the north pole, high latitude in North America, eastern Siberia, and Greenland sector. After the
vortex breakdown, the geographical extent of the positive ozone anomalies is mostly limited to
high latitudes with a semi-symmetrical shape in these cases. On the other hand, an elongated polar
vortex prior to the major SSW as seen in 2009 and 2018 (hereafter the elongated vortex SSWs) is
associated with negative ozone anomalies over a large extent of high latitudes, followed by
strongly positive TCO anomalies over an extensive area after vortex breakdown.

12        To investigate the connection of polar vortex strength and TCO, the scatter plot of the

zonally averaged EPV change at the potential temperature of 850 K versus the corresponding
change in TCO is shown in Figure 6. The ratio of change for each variable is estimated as the
average of 15 days after SSWs subtracted by the average of 15 days before the SSWs and divided
by the average of 15 days before the SSWs. A correlation between the magnitude of change in
EPV and TCO is evident. The elongated vortex SSWs (2009 and 2018) exhibit a higher magnitude
of change in both EPV and TCO in this period. It should be noted that making statistical
conclusions out of six SSWs is not the intention of this analysis, however, these observations
motivate further analysis of this study and future ones.

21        As the Greenland sector is one of the critical regions that exhibits positive ozone anomalies

before the displaced vortex SSWs, negative ozone anomalies before the elongated polar vortex,
and strong positive ozone anomalies after the vortex break down, the ozone variability over the
Greenland sector (60ºN to 80ºN and 10ºW to 70 ºW) as well as the zonal average (60ºN to 80ºN)
is analyzed to investigate the similarities and differences of the impacts of SSWs on zonal and
regional high latitude ozone. The structure of ozone anomalies in the zonal minus Greenland sector
is similar to the zonal average, therefore it is not included in further analysis. The Greenland sector





has been shown to be uniquely sensitive to dynamical forcing associated with the Quasi-Biennial
Oscillation (QBO) (Anstey and Shepherd, 2014; Bahramvash-Shams et al., 2019). Moreover, the
Greenland sector exhibits a very strong isolated stratospheric air circulation during wintertime, as
shown by the climatology of the polar vortex and its associated minimum temperature in Figure 1.
Thus, it is important to understanding the regional impact of SSWs on the Greenland sector.

6        To track the strength of the polar vortex, the average PV at the potential temperature of

850 K over the zonal average and the Greenland sector during 40 days before to 60 days after each
SSW is shown in the first column of Figure 7 The evolution of TCO over the zonal average and
the Greenland sector is shown in the second column of Figure 7. The climatologies of PV and TCO
for both the zonal average and Greenland sectors in Figure 7 are estimated based on non-SSWs
years between 2004 to 2019. To quantify the influence of SSWs on ozone, the average TCO for
the period spanning 40 days before to 60 days after the SSWs are shown in the bottom right of
each plot, as well as the ratio of the changes.

14       The climatological polar vortex position is located over the Greenland Sector (Figure 1)

and explains the higher intensity of climatological EPV over the Greenland sector compared to the
zonal climatology in Figure 7. The impact of minor SSWs in 2006 (around lag -25 and -19) and in
2008 (lag -30 and -15), as well as sudden polar vortex displacement to Eurasia in 2019 (lag -20)
showed a stronger signal on the averaged EPV over the Greenland sector by larger drop in EPV
compared to the zonal. The duration of the polar vortex recovery is defined by the number of days
between the date of the SSW and the date in which the zonal EPV returns to its climatological
value, as reported in the last column of Table 2. The fastest recovery of 30 days is observed in
2019 (also the weakest SSW) and the longest recovery duration of around 45 days is observed in
2009, 2013, and 2018. The recovery duration is similar with only a few days difference if the EPV
over the Greenland sector is used instead.

25        Compared to the 40-day average of TCO prior to the SSW, the highest percent zonal TCO

increase is observed for one of the elongated polar vortex SSWs with 29% in 2009. The maximum
percent TCO increases over the Greenland sector belong to both elongated polar vortex SSWs with



37% in 2018 and 31% in 2009. In all SSWs the magnitude of percent increased TCO is higher over
the Greenland sector compared to the zonal average with the exception of 2006, due to the effects
of the minor warming and a long period of high positive ozone anomalies over the Greenland
sector before the SSW.

5        The zonal TCO in half of the studied SSWs (2006, 2009, and 2013) does not get back to

the climatological TCO within two months after the SSWs, however, over the Greenland sector
the TCO reached the climatology in 30, 47, and 50 days respectively. In the other 3 SSWs the TCO
over Greenland returned to the climatology a few days before the zonal average. Therefore, on
average, the TCO over the Greenland sector shows a faster recovery and returns to climatological
levels within 25 to 55 days after the event as shown in Figure 7. No distinct pattern is observed in
elongated (2009, 2018) vs displaced vortex SSWs in regard to the recovery to the climatological
TCO.

13       Analyzing the vertical structure of ozone provides more details of the impact of SSWs.

Figure 8 shows the temporal evolution of the vertical structure of ozone as a cross-section of the
ozone anomalies for both the zonal average and the Greenland sector from 40 days before to 60
days after each SSW. The positive ozone anomaly in mid stratospheric layers (15 to 30km) starts
from a few weeks (15 to 25 days) prior to the displaced vortex SSWs (2006, 2008, 2013, and 2019)
over both the zonal average and the Greenland sector. The negative ozone anomalies 15 days
before the SSWs and extreme positive ozone after the SSWs in mid stratospheric layers for the
two elongated vortex SSWs (2009, 2018) are evident. The enduring impact of SSWs on ozone in
different atmospheric layers is clear in all cases and shows a similar pattern for both the zonal
averaged and the Greenland sector, as expected the structures of ozone anomalies are smoother in
the zonal average compared to the Greenland sector compared as shown in Figure 8. More intense
and frequent negative ozone anomalies in mid stratospheric layers at the end of study period is
also reflected by the faster recovery of TCO to climatological values as shown in Figure 7. The
shortest impact on TCO belongs to 2008 because of multiple disturbances in the circulation and
the shortest duration of easterlies (Table 2).





To highlight the temperature variation, Figure 9 shows the cross-section of the temperature
anomaly for the zonal average from 40 days before to 60 days after each SSW. Figure 9 focuses
only on the zonal average, as the anomaly of temperature profile had similar patterns over the
zonal and the Greenland sectors. The positive temperature anomalies in mid stratospheric layers
start a few weeks before the SSWs in the 4 cases of a displaced vortex (2006, 2008, 2013, and
2019). On the other hand, the intrusion of the positive temperature anomalies to mid stratospheric
layers is almost coincident with SSWs in the 2 elongated vortex cases. The duration of positive
temperature anomalies in mid stratospheric layers is 10 days to 30 days shorter than ozone positive
anomalies (Figure 8 and Figure 9). The positive temperature anomaly is more persistent at lower
levels of the stratosphere, where the enduring impact of SSWs on mid-stratosphere ozone (up to
25 -30 km) is clear in all of the SSWs studied here.
**6. Discussion**
The cyclonic polar vortex during wintertime is generated in response to the seasonality of
radiative cooling. The intensified wave forcing before the SSW is manifested by both accelerated
tropical upwelling and polar downwelling, which leads to advection of low EPV air parcels
poleward. The conservation of EPV causes anticyclonic circulation, which gradually drives
easterly zonal mean zonal winds, and leads to displacement or splitting of the polar vortex. The
resultant reduction in the vorticity induces strong descent and consequently an adiabatic
temperature increase in the stratosphere (Matsuno, 1971; Limpasuvan et al., 2012).
Here the MERRA-2 dataset is used to determine the impact of the dynamical terms during
each SSW. Because of the constraints in tracer continuity estimation using equation (2), these
analyses are estimated over the Arctic zonal average only and not the Greenland sector. The
vertical component of the residual circulation ($\overline{w}^*$) as defined in equation (3) is an indicator of
wave forcing. The cross-section of the vertical component of residual circulation during 40 days
prior to and 60 days after the SSW over the zonal average (60ºN to 80ºN) is shown in Figure 10.
More intense downward propagation is shown as darker blue. The increased wave forcing
preceding the SSW is evident in Figure 10 with negative $\overline{w}^*$ anomalies, which indicate strong



downwelling in the zonal average. Occurrences of minor SSWs are evident by the early appearance
of increased wave forcing, as seen in 2006 and 2008. A very intense and abrupt increase in
downward propagation was observed in 2009. Disturbed circulations in the middle stratosphere
before the SSWs are seen in 2018 and 2019 (lag -30 to -20).
Following the SSW, residual circulation is weakened as shown in Figure 10. The intensity
of increased wave activity is reduced shortly after the SSW.  However, the decrease in wave
activity is gradual, in general, and lasts a few weeks as shown in Figure 10. The suppressed wave
activity creates preferable conditions for the recovery of the zonal mean zonal wind, temperature,
and ozone. Shortly after the SSW, the recovery starts in the upper stratosphere as shown in Figure
9. However, different radiative relaxation time scales cause a slower recovery in the lower
stratosphere compared to upper stratospheric layers (Dickinson, 1973; Randel et al., 2002;
Hitchcock and Simpson, 2014). The dynamical alteration suppresses any further upward
propagation of the planetary waves, which explains the descending pattern of temperature up to
weeks after the SSW (Matsuno, 1971).
The impact of each term in tracer continuity (equation (2)) on ozone for each SSW are
investigated and shown in Figure 11. The composite effect of chemistry during SSWs is important
in the upper stratosphere (de la Cámara et al 2018b). The analysis of dynamical parameters in this
study are limited to 30 km to minimize the impact of chemical processes. The cross-section of
ozone tendency ($dO_3/dt$, left side of equation (2)), the horizontal component of eddy mixing
$e^{(z/H)}(M_y/dy)$ ($M_y$ as defined in equation(4) ), the vertical component of eddy mixing $e^{(z/H)}(M_z/dz)$
($M_z$ as defined in equation(5) ), the horizontal advection transport (the first term on the right side
of equation (2)), vertical advection transport (the second term on the right side of equation (2)),
and summation of right side equation(2) (called the estimated ozone tendency) during the 40 days
prior to and 60 days after the SSW over the zonal average are shown in Figure 11.
The estimated ozone tendency (last column of Figure 11) shows that using MERRA-2
fields, dynamical terms of tracer continuity can simulate the main features of the observed ozone
tendency (first column of Figure 11) below 30 km and use these estimates to investigate the impact





of different terms of tracer continuity on ozone. The key role of vertical advection and horizontal
eddy mixing on ozone tendency is evident in Figures 11. Vertical advection is the main driver of
ozone tendency in the mid stratosphere. Intensified residual circulation (Figure 10) dramatically
impacts the ozone increase. A significant signal of vertical advection is evident from 10 to 30 km
in all six SSWs and is coincident with enhanced wave activity (Figure 10), which is magnified
around SSWs; however, it persists well after the vertical residual circulation signal disappears, up
to two months after the SSWs. The sudden and intensified vertical advection is more magnified in
2009 and 2018 with an enduring elongated polar vortex.

9       Horizontal eddy mixing is the second important contributor in ozone tendency over the mid

stratosphere. While vertical advection builds up the ozone tendency, horizontal mixing tends to
balance and weaken the ozone tendency. Increased wave activity and large-scale mixing influence
a prolonged enhancement of the diffusivity of PV flux, which leads to increased horizontal eddy
transport (Nakamura, 1996; de la Cámara et al., 2018a; 2018b). Vertical eddy mixing has a clear
signal above 20 km during minor and major SSWs. Horizontal advection has the least significant
contribution to ozone tendency. The dominant contribution of vertical advection on mid-
stratospheric ozone variability (15 to 30 km) using MERRA-2 dynamic parameters are consistent
with climate model analysis (Tao et al., 2015; de la Cámara et al., 2018b). Considering the larger
uncertainties of ozone estimation in MERRA-2 below 15 km, and the possibility of larger
uncertainties in dynamic parameter estimations, this study does not analyze the impact of the
dynamics on ozone in the lower stratosphere. Using climate models, it has been shown that the
horizontal eddy mixing term plays a key role in the ozone variability in the lower stratosphere
during the SSWs (de la Cámara et al., 2018b).
The time series of vertically integrated (15 to 30 km) ozone tendency, horizontal eddy
mixing, vertical advection, and the residual of tracer continuity considering all terms in equation
(2) are shown in Figure 12. The major contribution of vertical advection on ozone tendency is
evident in Figure 12. The higher intensity of ozone tendency and vertical advection and their strong
correlation coincident with the SSW date of the elongated polar vortex (2009 and 2018) stand out
in Figure 12.





Although the estimated ozone tendency (last column in Figure 11) simulates most features
of the observed ozone tendency (the first column in Figure 11), they are not identical. The
vertically integrated difference in observed and estimated ozone tendency is shown as the residual.
The residual of tracer continuity results from both the numerical approximation of terms in
equation (2) (errors in the vertical derivatives over high latitude can be large as $\cos(\varphi)$ gets small)
as well as the uncertainties in the balance of dynamical parameters in the reanalysis due to the data
assimilation process (Martineau et al. 2018). Also, the possibility of chemical processes during
splitting or displacement of the polar vortex out of the polar night region might contribute to the
residual of tracer continuity. It should be noted that when viewing individual events, the plots are
expected to be noisier than the average of numerous events.
**7. Conclusion**
SSWs are a major manifestation of disturbed stratospheric circulations. The altered
dynamics influence the cycle of trace gases including ozone. MERRA-2 reanalysis is used to
investigate the influence of six recent SSWs on ozone for the zonal average at high latitudes (60ºN
to 80ºN).
The comparison of the MERRA-2 ozone dataset with a unique density of observations at
high latitudes from 2004 to 2020 provides an update to previous evaluations and provides
understanding of the performance of MERRA-2 during high variability associated with extreme
dynamical events such as SSWs. Comparisons are applied during December to May for each SSW.
MERRA-2 shows good agreement with ozonesondes and FTIR observations in the middle
stratosphere during highly altered dynamics of SSWs.
Comparison with ozonesondes at three high latitude locations showed the mean difference
ratio of 3% (±7%) in the stratosphere layer (10-30 km). However, the uncertainties are larger from
the ground to 10 km. From 5km to 10km, a non-significant (higher standard deviation) negative
mean bias exists in all sites (-8% to 15%). The highest standard deviation of difference ratios is





observed at G-5 km (<20%). A positive bias is observed at surface levels where observations show
depleted ozone due to bromine reactions.
Using a smoothing method, MERRA-2 is compared to five NDACC FTIR sites in four
vertical layers (ground-8km, 8-15km, 15-22km, and 22-30km) during SSWs. These layers are
defined based on the sensitivity of FTIR sensors. Overall, higher uncertainties are observed at the
lowest level with 18% std. The best agreement is observed between 15-22 km and 22-34 km with
-2%($\pm$5%) and -4%($\pm$5%) mean(std) difference ratios. These results emphasize the high quality of
MERRA-2 and motivate its usage in mid stratospheric ozone analysis at high northern latitudes
during highly disturbed dynamical events. Higher uncertainties in UTLS are also expected because
MLS has lower sensitivity at lower altitudes and the dominant contribution of MLS in MERRA-2
reanalysis. Moreover, this study emphasizes the importance of independent ozone observations,
such as ozonesondes and FTIR retrievals, as a means to evaluate models and assimilation
estimations around the globe.
Using the MERRA-2 dataset, the variability of ozone during the SSWs and associated
dynamic parameters are investigated. The evolution of the polar vortex and its impact on the ozone
variability is studied using the average EPV at the potential temperature of 850 K. We identify two
different patterns in the polar vortex before the SSWs and the subsequent impact on ozone. In 2009
and 2018, an elongated polar vortex is observed before the SSWs which caused a dominant-
negative ozone anomaly at northern high latitudes and is followed by an extensive positive ozone
anomaly with large geographical extent. The TCO increases rates and the magnitude of change in
EPV after these cases are large and the intrusion of positive temperature anomalies to the mid
stratosphere is coincident with SSWs date.
During the SSWs in 2006, 2008, 2013, and 2019, the polar vortex is displaced towards
Europe, and the TCO exhibits positive anomalies before the SSWs in a large geographical region
of northern high latitudes (outside the polar vortex). The positive TCO anomalies after the SSW
have a smaller extent, and the magnitude of TCO variability and EPV change is smaller compared



to observed changes during the elongated vortex event as seen in 2009 and 2018. The positive
temperature anomalies in the middle stratosphere appear a few weeks before the SSW.

3        A strong relation is observed between the magnitude of change in the averaged EPV 15

days after compared to 15 days before the SSW and the magnitude of TCO change for the same
period for all six studied SSWs.

6        The Greenland sector is one of the critical regions that is impacted by negative TCO before

the elongated polar vortex in 2009 and 2018; positive TCO occurs before displaced SSWs. To
identify the similarities and differences of zonal versus the regional impact of SSWs on ozone, the
analyses are applied over the Greenland sector as well as the zonal average. The general structure
of the vertical ozone anomaly over the Greenland sector is similar to the zonal. However, as
expected the ozone anomaly over the zonal average is smoother than the Greenland sector which
results in a more magnified TCO increase over Greenland. The increased rate over the Greenland
sector is between 15% in 2006 to 38% in 2018, while the zonal average ranges between 8% in
2008 to 29% in 2009. Moreover, TCO exhibits a faster recovery to the climatology values over
this region compared to the zonal average.

16        We examined the dynamical terms associated with ozone tendency and investigated the

evolution of ozone variability for each SSW using MERRA-2. The main features of observed
ozone tendency are captured by the dynamical terms of the tracer continuity equation up to 30 km
using MERRA-2 variables. Vertical advection is shown to be the main contributor of ozone
tendency in the middle stratosphere during the SSWs and is more magnified during the enduring
elongated polar vortex in 2009 and 2018. The impact of vertical advection coincides with the time
of enhanced wave activity but can persist up to two months after the SSWs.

23        Suppressed wave activity initiates the recovery of temperature and ozone. However, the

upper stratosphere experiences a faster recovery compared to the lower stratosphere because of the
different radiative relaxation time scales. The faster recovery of zonal temperature and ozone at
middle stratosphere within 30 days is recorded for 2008 with the shortest duration easterly zonal
mean zonal winds. The zonal TCO did not return to climatological TCO values within two months



1 after the SSWs in 2006, 2009, and 2013. The positive ozone anomaly in the middle stratosphere

2 lasts longer than the positive temperature anomaly in most of the SSWs by 10 days or more.

3   In conclusion, the MERRA-2 dataset is shown to capture ozone variability in the middle

4 stratosphere and provides dynamical information to investigate the impact of SSWs. The impact

5 of SSWs on ozone and the role of vertical advection is shown to be more intense in 2009 and 2018

6 with an elongated polar vortex compared to displaced vortex in 2006, 2008, 2013, and 2018. The

7 magnitude of change in ozone is correlated with the magnitude of EPV change during the SSWs.

8 Using a more extended dataset could help to shed light on further details and to create more robust

9 statistics regarding Arctic SSWs. Considering the impact of high latitude ozone on global climate

10 and lower latitude surface temperature, the dramatic ozone increases over high latitudes during

11 SSWs points to consequences of these events on the global earth system and possible

12 environmental/ecosystem changes that could be investigated in future studies.

13 **Acknowledgements**

   We acknowledge NASA's Global Monitoring and Assimilation Office (GMAO) for providing the Modern-Era Retrospective analysis for Research and Applications, Version 2 (MERRA-2). We acknowledge the Ozone and Water Vapor Group at the Earth System Research Laboratory of the National Oceanic and Atmospheric Administration for use of the ozonesonde data and the science technicians at Summit Station, Greenland for launching the ozonesondes. We acknowledge World Ozone and Ultraviolet Radiation Data Centre (WOUDC) for providing Canadian and Norwegian ozonesondes. We acknowledge the Network for the Detection of Atmospheric Composition Change (NDACC) for providing trace gas retrievals from solar FTIRs. This research was supported by NSF grants PLR-1420932 and PLR-1414314.

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





**Table 1.** Site locations for NDACC FTIRs and ozonesondes. Uncertainties of FTIRs at three sites with ozonesondes are given by averaged subtraction and standard deviation of ozonesondes from the retrieved ozone from FTIR.

| station | Longitude | Latitude | Solar FTIR Time period | Ozonesonde Availability period | %full PCO uncertainties | %Uncertainties 10 km -30km |
|---|---|---|---|---|---|---|
| Eureka | 274 | 80 | 2006-now | 1992-now | 7% +/- 7% | 1% +/- 7% |
| Ny-Ålesund | 12 | 79 | 1995-now | 1992-now | 2% +/- 4% | 7% +/- 8% |
| Thule | 291 | 77 | 1999-now | 1991-2016 (very sparse) | 3% +/- 6% | 3% +/- 6% |
| Summit Station | 39 | 72 | - | 2005-2017 | | |
| Harestua | 11 | 60 | 2009-now | - | - | |
| Kiruna | 20 | 68 | 1997-now | - | - | |




**Table 2.** SSWs dates, duration, magnitude, and the duration of polar vortex from 2004 to 2020. The number of easterly days at 10 hPa over 60 N is shown as the duration SSW. The magnitude of SSWs is defined by the minimum zonal-mean zonal wind at 10hPa over 60 N during each SSW. The total number of easterly days associated with the event is not necessarily consecutive. The duration of polar vortex recovery is defined as the number of days that the zonal averaged EPV takes to reach the climatological zonal EPV.

| SSWs date | Number of easterly days at 10 hPa over 60 N | Minimum zonal-mean zonal wind at 10hPa over 60°N (m/s) | Vortex recovery (days) |
|---|---|---|---|
| 21 Jan 2006 | 26 | -26 | 36 |
| 22 Feb 2008 | 15 | -15 | 35 |
| 24 Jan 2009 | 30 | -29 | 45 |
| 6 Jan 2013 | 22 | -13 | 45 |
| 12 Feb 2018 | 19 | -24 | 45 |
| 2 Jan 2019 | 21 | -10 | 30 |


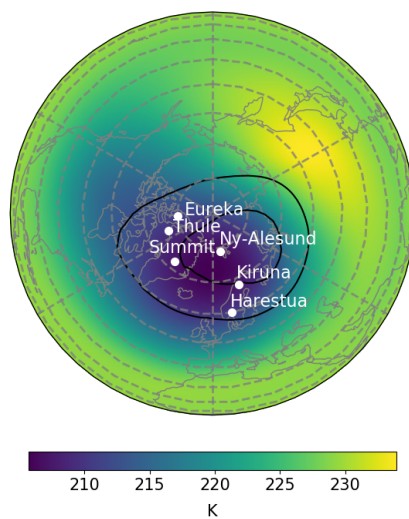

**Figure 1.** The climatology of temperature at 10 hPa and potential vorticity (PV) at the potential temperature of 850 K during wintertime (DJF) over the northern hemisphere. The climatology is based on non-SSW years from 2004 to 2019. The map coloring shows the average winter temperature. The black contour lines are 60 and 80 PV units ($10^5$ K $m^2$ $Kg^{-1}$ $s^{-1}$). The locations of the observational sites are shown as white dots.

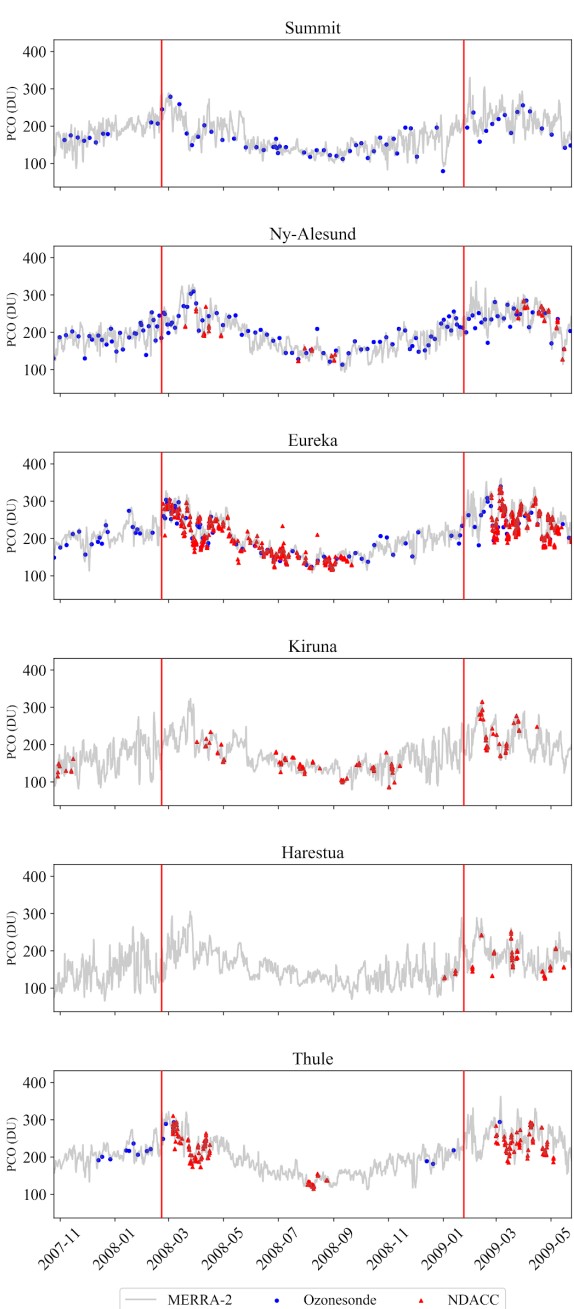

**Figure 2.** Time series of 3 hourly partial column ozone (PCO) of ground to 20 km derived from MERRA-2, solar FTIR, and ozonesondes at the study sites from winter 2007 to spring 2009. MERRA-2 is shown as the gray line. NDACC FTIR data and ozonesondes are shown as red triangles and blue circles, respectively.



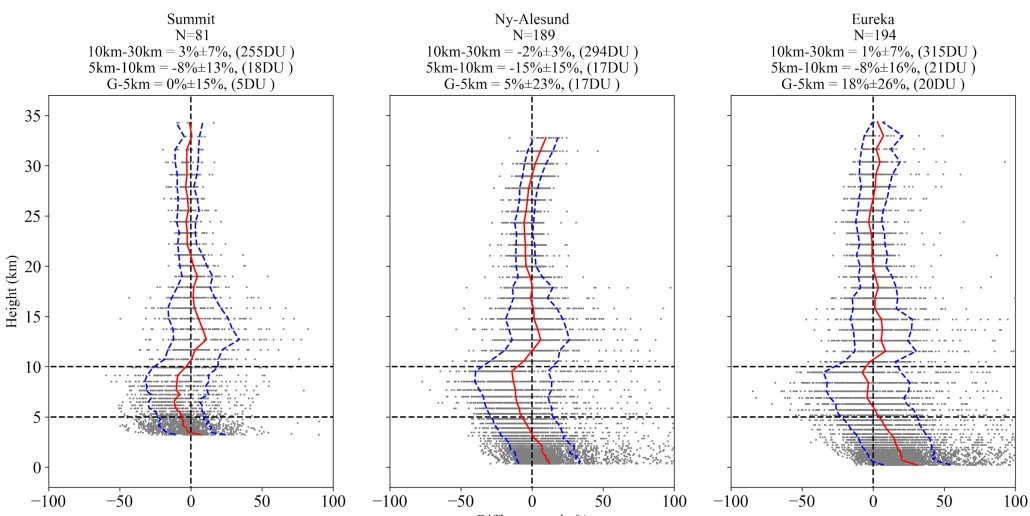

**Figure 3.** Difference ratios of ozonesonde and MERRA-2 at each layer at three sites from 1 Dec to 1 May for six year of SSWs. The difference ratio is the subtraction of ozonesonde from MERRA-2 ozone dataset divided by ozonesonde for each layer. The mean difference ratio is shown as the red line. The standard deviation of the difference ratio from the mean is shown at the blue line. The number of coincident ozonesonde and MERRA-2 comparisons between 1 Dec and 1 May for the six years of SSWs (N) is shown under each site name. The mean and standard deviation of PCO difference for 3 layers: 10km-30km, 5km-10km, Ground-5km are summarized for each site. The average PCO value for each layer is shown in parentheses.



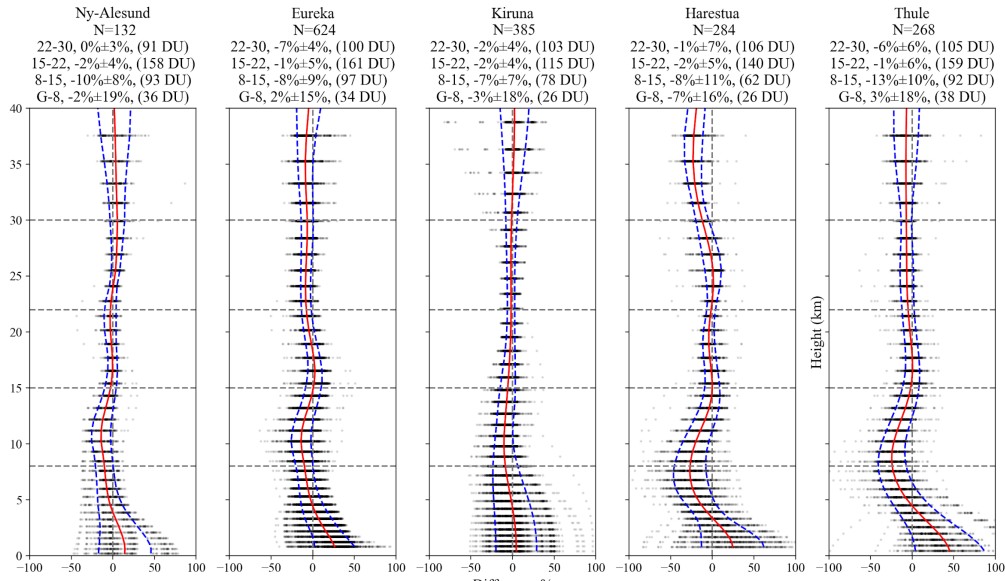

**Figure 4.** Difference ratios of FTIR retrieved ozone from MERRA-2. For each layer at each station, the mean ± standard deviation of NDACC retrieved ozone is subtracted from MERRA-2 and divided by retrieved value. The mean and standard deviation of difference ratios for each layer are shown as the red and blue lines. Statistical summaries of the MERRA-2 and NDACC comparisons in four layers of ground to 8 km, 8km-15km, 15km -22km, and 22km- 30km for each station are shown on top of each plot. Mean and standard ratio difference and the average PCO of each layer are shown for each site.



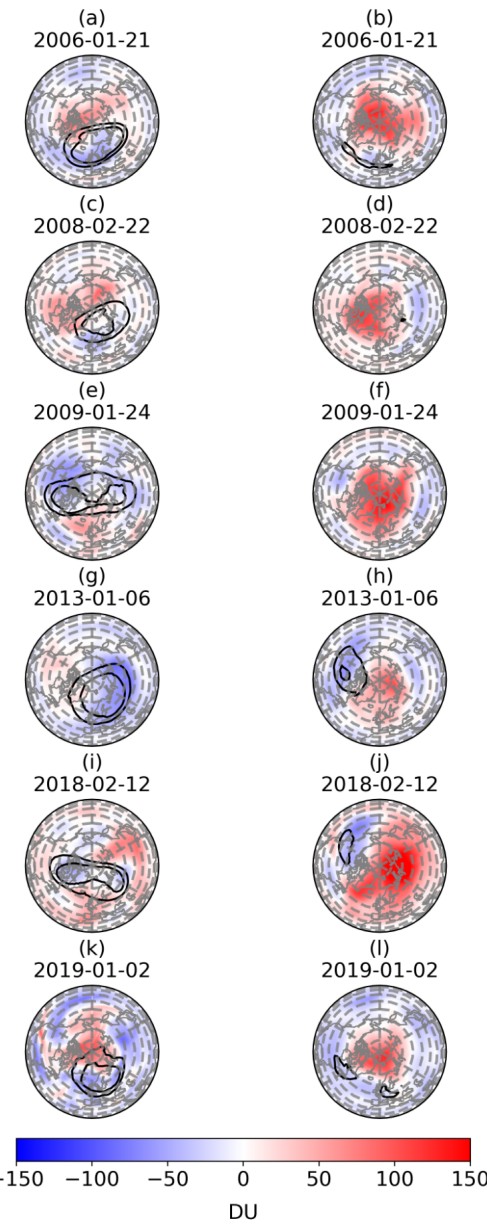

**Figure 5.** TCO anomaly using 15 days prior (first column) and 15 days after to each SSW (second column). PEV at the potential temperature of 850k is averaged for the same period similar to TCO. Contour lines show the EPV map at 60 and 80 $10^5$ K m$^2$ Kg$^{-1}$ s$^{-1}$.





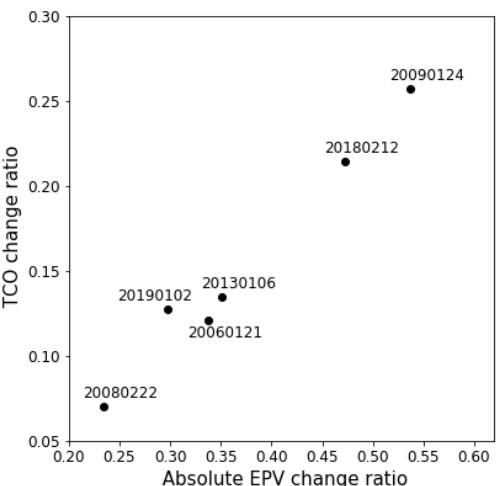

**Figure 6.** The zonally averaged EPV change ratio at the potential temperature of 850 K against the corresponding change in TCO for each SSW. The ratio of change for each variable is estimated as the average of 15 days after SSWs subtracted by the average of 15 days before the SSWs and divided by the average of 15 days before.

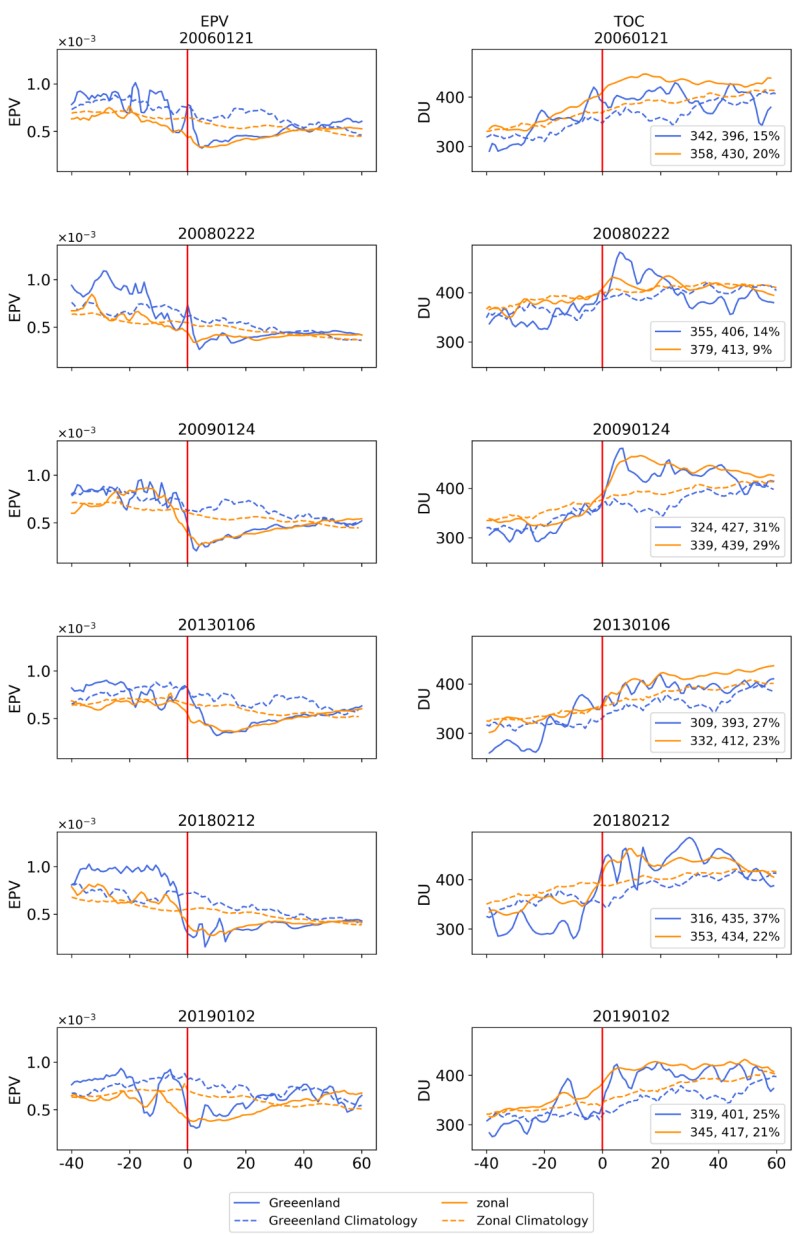

**Figure 7.** EPV at the potential temperature of 850k (first column) and TCO (second column) over the Arctic zonal mean 60-80N (orange line) and Greenland sector (blue line) during 40 days before and 60 days after each SSWs (each row). Climatology of EPV and TCO for the zonal and Greenland sector are shown in orange and blue dashed lines, respectively. The average Total Column ozone (TCO) during 40 days before and 60 days after, and the percentage of change for each SSWs are shown in the bottom corner of the second column.

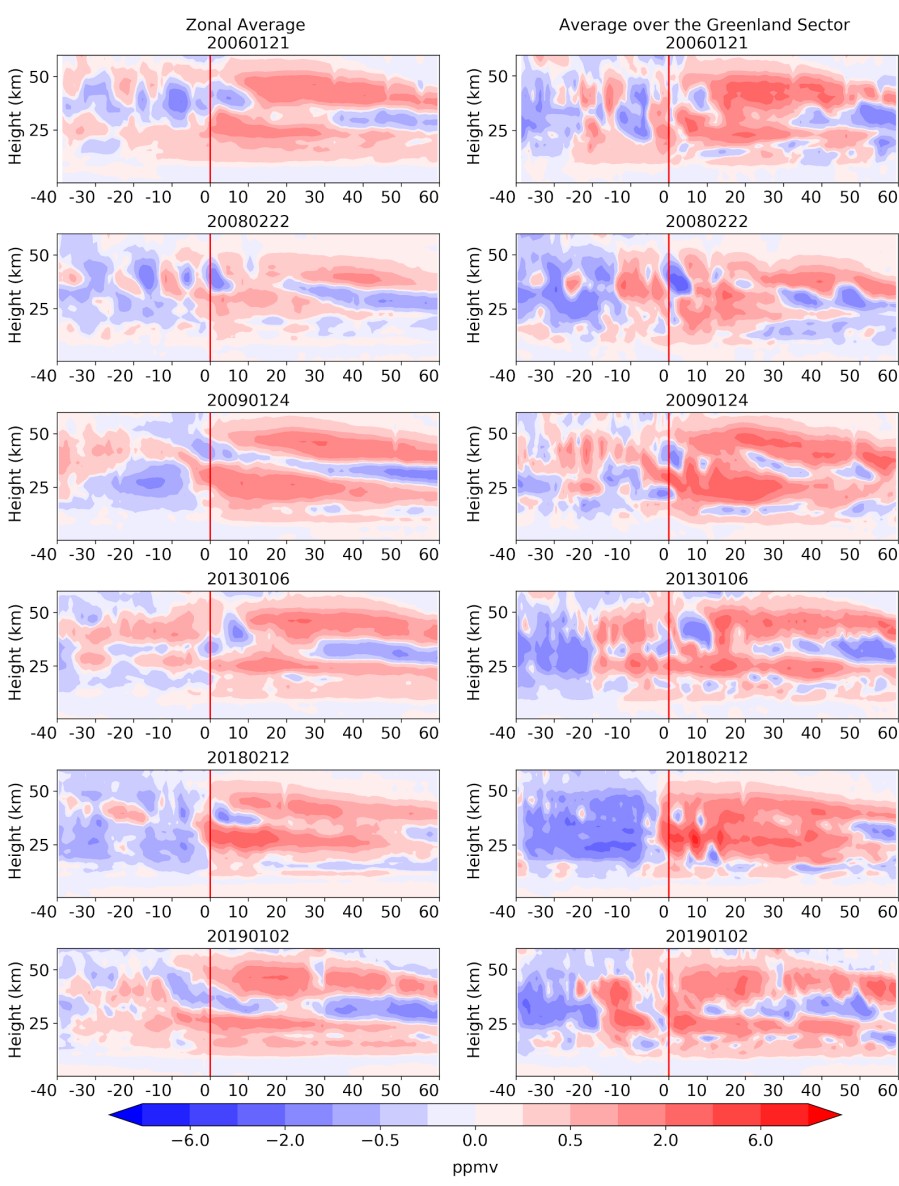

**Figure 8.** The cross section of ozone anomaly during 40 days before to 60 day of each SSWs averaged over the zonal averaged and Greenland sector. The vertical red line shows the SSWs incident date. Climatology was created using non-SSWs years since 2004. The vertical coordinate is the log-pressure height.



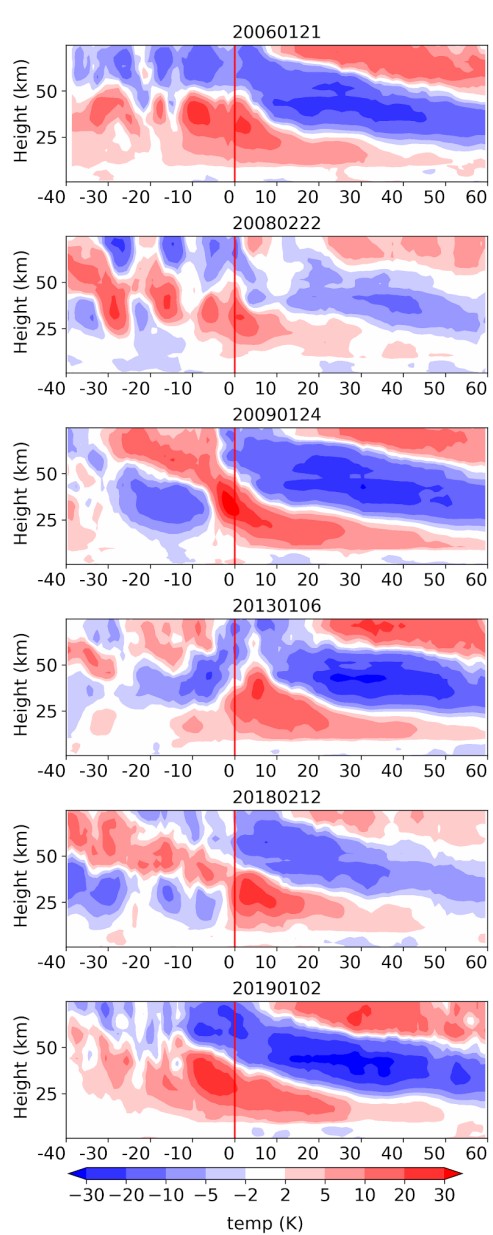

**Figure 9**. Similar to figure 8 but for the temperature anomaly for zonal average.

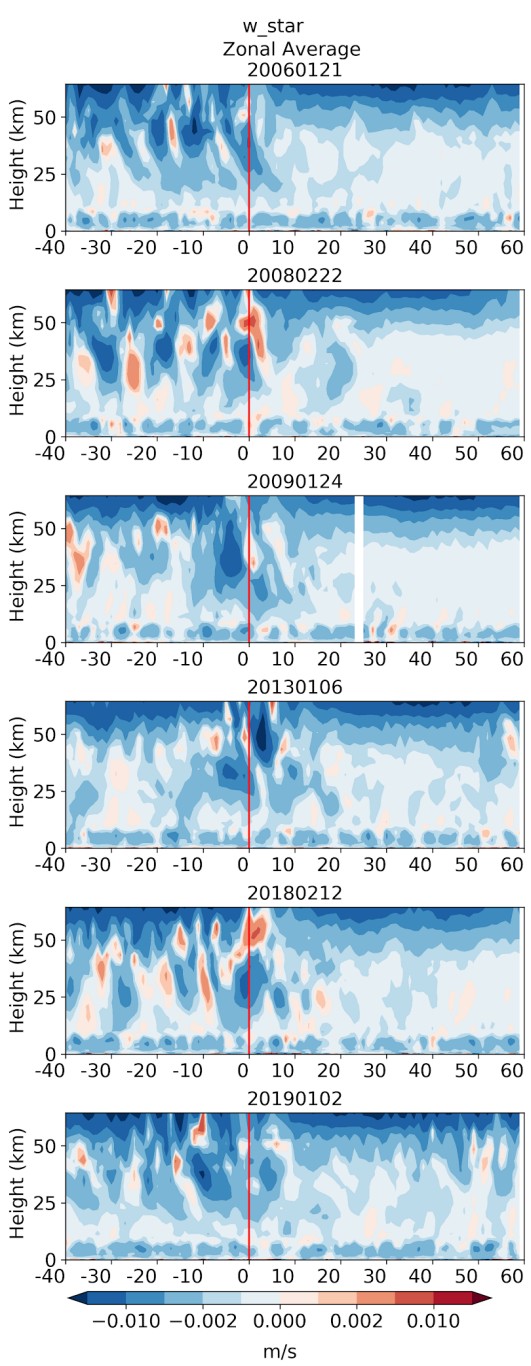

**Figure 10.** Similar to figure 8 but for the of the vertical component of the residual circulation,$\overline{w}^*$, for zonal average.

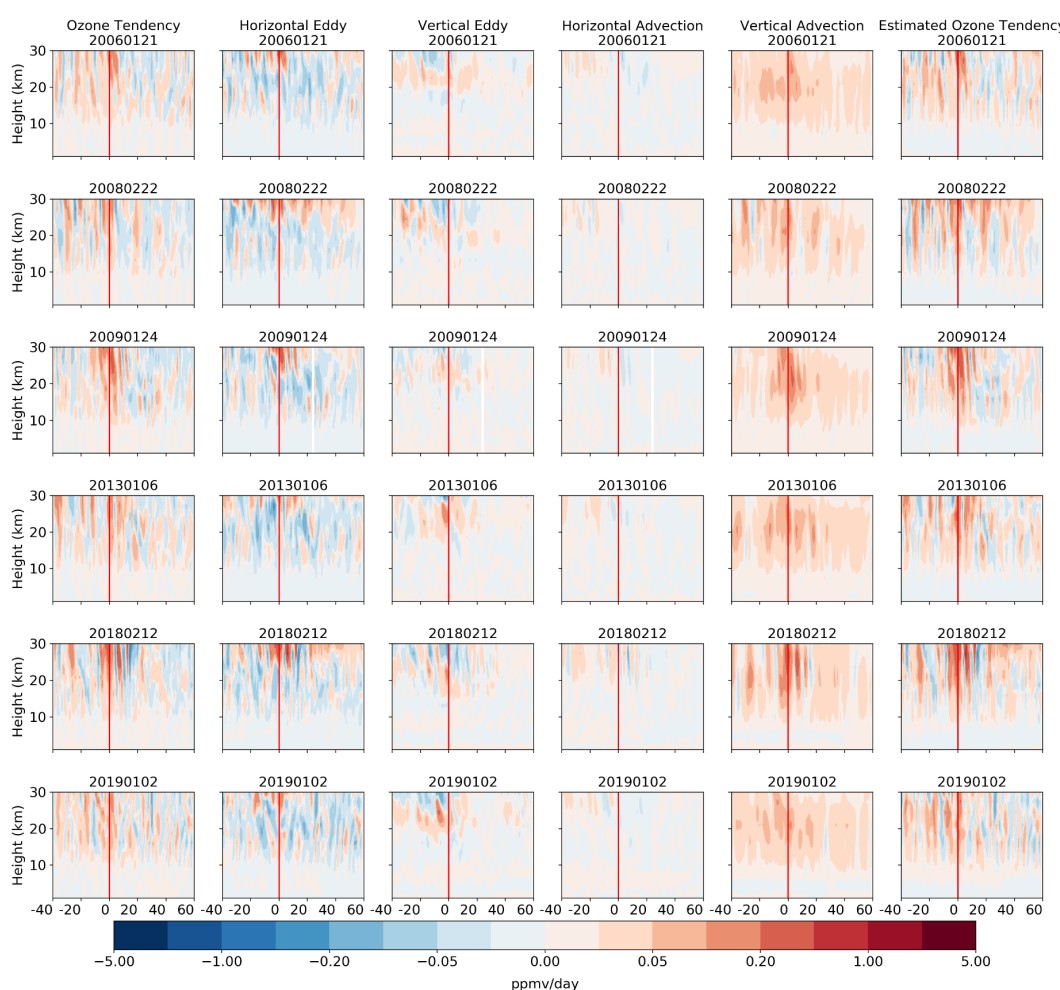

**Figure 11.** Same as Figure 8 for ozone tendency, horizontal and vertical component of eddy mixing, and horizontal ($-\overline{v}^* \, \overline{x}_y$) and vertical ($-\overline{w}^* \, \overline{x}_z$) component of mean advection, and the indirect ozone tendency using the right-hand side of equation (2). Summing four middle columns leads to the estimated ozone tendency on the sixth column. The vertical axis is the log-pressure height. The y axis is limited to 30 km to minimize the impact involved chemical processes on ozone evolution during SSWs.





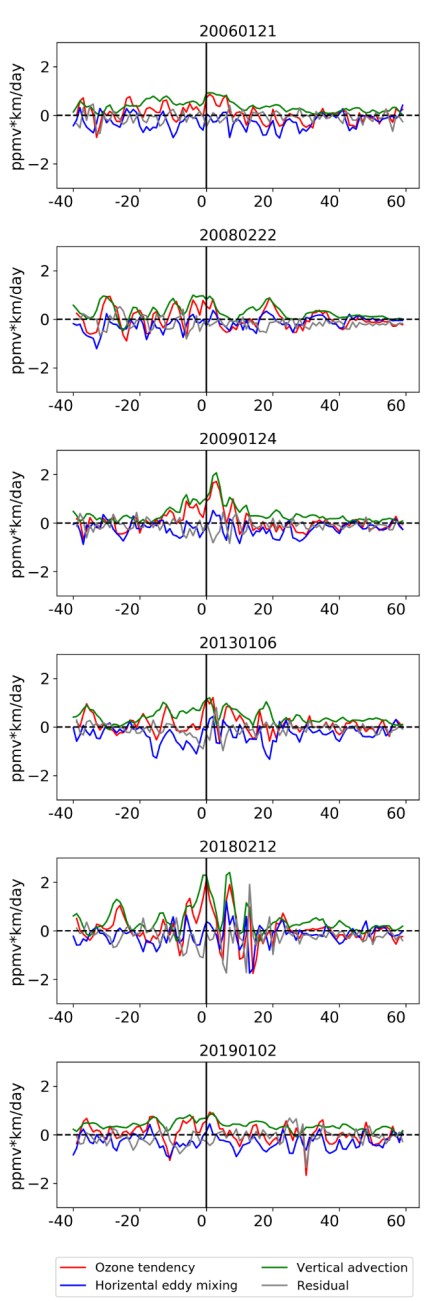

**Figure 12.** Time series of vertically integrated major elements of tracer continuity equation 2 from 15km to 30 km (Andrews et al, 1987). Ozone tendency is shown as the red line. The horizontal component of eddy mixing, $e^{(z/H)}dM_y/d_y$, is shown in blue line, the vertical component of vertical advection, $-\overline{w}^* \bar{x}_z$, is shown in the green line. The residual of all elements of tracer continuity is shown in the gray line.