# Peer review of "Analyzing ozone variations and uncertainties at high latitudes during Sudden Stratospheric Warming events using MERRA-2"

_Atmospheric Chemistry and Physics, 2021_

## Referee Comment (RC1)

**Review of the manuscript *"Analyzing ozone variations and uncertainties at high latitudes during Sudden Stratospheric Warming events using MERRA-2"* by Shima Bahramvash Shams et al.**

This study examines the evolution of polar ozone during six recent major sudden stratospheric warming events (SSW) using the MERRA-2 reanalysis. The analysis is preceded by evaluation of the MERRA-2 ozone using independent observations and focusing on the region and periods of interest, which provides a nice complement to previous validation studies that looked mainly at the global picture. The authors find that the impacts of SSWs on ozone were largest in 2009 and 2018 ("elongated vortex" cases) compared to the other events ("displaced vortex") and identify vertical advection as the key mechanism responsible for the formation of these positive ozone anomalies during all the SSWs analyzed in this study.

This paper is interesting and certainly suitable for publication in ACP subject to some minor revisions as delineated below. It's nice to see a detailed study of a number of recent SSWs in one place along with an ozone budget analysis for each of them. It certainly adds to our understanding of the role of transport during these events. It's also great to see another paper that demonstrates the value of reanalysis ozone data. It is only recently that our community began to take advantage of the global coverage, high resolution and dynamical consistency that these products offer. The additional validation of MERRA-2 ozone, focused on the region of interest is especially valuable. While I ended up having a rather large number of comments and suggestions, none of them are serious objections, and I believe all of them can be easily addressed. The most important ones concern the need to highlight the novel aspects of this work in the context of other similar studies (some of them not cited) (general comment #1), the choice of these particular six events and a possibility of adding 2021 to the list (general comment #2), and terminology (#3). I hope my comments will be useful.

**General comments**

1.  Can you place your results in the context of previous papers discussing the role of dynamical ozone resupply, e.g., Tegtmeier et al., 2008; Strahan et al. (2016)? It would help if you clearly delineated the novel aspects of your study against the backdrop of the existing literature of the subject.
2.  Butler et al. (2017) as well as the SSW Compendium (https://csl.noaa.gov/groups/csl8/sswcompendium/majorevents.html) list several more SSWs than those discussed in the paper: 2007, 2008, and 2010. I assume there's a good reason for the selection discussed in this study to be what it is, but it needs to be explained, especially since the definition of SSW used here is the same as that applied in the Compendium. In addition, ideally, I would like to see 2021 added to the analysis. Seven events are better than six. Note that it would likely be the first paper that talks about ozone during the 2021 SSW.
3.  There is a terminological confusion regarding the use of the word "model" where you really mean either data assimilation system, assimilated data, or reanalysis. I tried to catch those instances in my specific comments below. I think it's important to remember that a reanalysis is **not** a model simulation. Rather, it is fundamentally a data-driven product. Calling a reanalysis a "model" is equivalent to calling a retrieved satellite data set "a

radiative transfer model". The role of the general circulation model in a data assimilation system is to propagate information from observations in time and space over a period of several hours. That's all it does. Things do get a bit muddled in places/periods devoid of observations, but your study looks at regions well constrained by data, so that's not an issue here.

4. You use the terms lower and middle stratosphere somewhat loosely. Please define them somewhere in the methods section.

5. You choose to exclude the lower stratosphere from this analysis citing the high uncertainty of MERRA-2 ozone there. While it's completely OK to make that choice I have a problem with the stated motivation. It's certainly true that the uncertainties get larger closer to the tropopause, but I don't think one can say, based on the validation that you did or the results in Wargan et al (2017), that there's no useful information down there, i.e. that the true variability is completely obscured by uncertainty. The ozonesonde comparisons in Fig. 3 suggest difference standard deviations of about 25%. Below I plot the time series of MERRA-2 ozone at Eureka at 150 hPa along with 25% and 50% envelopes. Clearly, the dynamical variability around the 2009 SSW is still very much discernible in the sense that the magnitude of the large jumps exceeds the uncertainty. Additionally, note that Albers et al. (2018) found useful ozone information in MERRA-2 in the lowermost stratosphere and Knowland et al. (2017) as well as Jaeglé et al. (2017) successfully studied stratospheric intrusions into the troposphere using MERRA-2 ozone. Again, I agree that any results in the LS would have larger uncertainty, but I can't agree that they would be worthless, and the present wording in the paper seems to imply that. I'm not suggesting extending the analysis. It's just a matter of phrasing things in a more nuanced way.

[Figure]

6. In several places the language of the paper implies (or even outright) states causal relationships where causality is not immediately demonstrated. I provide some examples in my specific comments. There are sentences like "*the vortex displacement toward the southeast (Europe) prior to the 4 major SSW as seen in 2006, 2008, 2013, and 2019 (hereafter the displaced vortex SSWs) caused an early positive ozone anomaly*", "*an elongated polar vortex is observed before the SSWs which caused a dominant negative ozone anomaly*", "*SSWs and their impact on ozone*" (the title of a section). I get it that Figures 11 and 12 and the discussion in Section 6 do provide evidence for a causal mechanism, but the reader doesn't know that in advance. I suggest rephrasing these causal

statements as something more "relational" (Fig 6 and the wording on P24 L3-5 are very good) and explaining that a dynamical mechanism will be elucidated in Section 6. You can then state your results in causal terms in the conclusions as you do. It's really about streamlining the argument.

**Specific comments**

P2. L19-22. I believe that this "ozone gets made in the tropics and transported to the high latitudes" description is simplified to the point of being incorrect (albeit quite commonly used). Please, take a look at the discussion around Fig. 5.11 in Bresseur & Solomon 2005 (in my edition it starts on page 286 with "It is sometimes stated that ozone is produced where its mixing ratio maximizes…"). This does not affect anything in your paper of course.

P2. L24-25. Would it make sense to cite the recent review paper on SSWs by Baldwin et al., 2021?

P4. L3-4. "numerical and assimilation models". I struggle with this terminology. I understand that by "numerical models" you mean general circulation models or numerical weather prediction models. I can live with the short-hand version "numerical models". But I think the word model should not be applied to data assimilation systems, which comprise a model component and a statistical analysis scheme. I suggest "numerical models and data assimilation systems". See my general comment #3.

P4. L15-16. At first read this sentence confused me: why focus on zonal structures? SSWs are very non-zonal! But you're actually doing much more than that: the Greenland sector, polar averages, maps. I suggest rephrasing it.

P4. L17. "assimilation model". See above. Why not just say reanalysis?

P5. L7-9. The description of MERRA-2 needs some rewriting. It's not clear to me what it means that there's a variety of models incorporated in MERRA-2. Do you mean GCM, land model, parameterized chemistry, etc.? There aren't multiple "general circulation model**s**" in it as the text implies. There is one. Also, I don't know what you mean by "extended reanalysis". No other reanalysis is incorporated in MERRA-2. If you mean something like OSTIA (the SST data set that provides boundary conditions for the GCM), it's not a reanalysis, at least not in the same sense as MERRA-2.

P5. L16. I wouldn't say that it's been extensively used in trend studies. To my knowledge only Wargan et al. (2018) derived trends from (suitably corrected for discontinuities) MERRA-2 (there was a follow-on paper by Orbe et al. 2020, but MERRA-2 ozone is sort of tangential there). Also, please consider adding a very interesting study by Albers et al. (2018) to these citations.

P5. L21. Please, provide a citation for the data collection used in this study. Information on how to cite the MERRA-2 pressure-level output is here: https://disc.gsfc.nasa.gov/datasets/M2I3NPASM_5.12.4/summary under "data citation". For the

model-level output: https://disc.gsfc.nasa.gov/datasets/M2I3NVASM_5.12.4/summary (I'm providing both in case I misunderstood which one you used; see the next comment).

P5. L21. Why use pressure-level output if much higher vertical resolution model-level output is available? But P8 L16 talks about using the model levels. So which collection is really used?

P5. I think it should be mentioned that MERRA-2 assimilates MLS ozone down to 177 hPa between 2004 and 2015, then switches to version 4.2 retrievals going down to 215 hPa. This version (and vertical range switch) results in some differences between the pre- and post 2015 periods, whereby the latter has likely more accurate ozone, especially in the lower stratosphere.

P5. L22. "Assimilated/reanalysis models". Why not just say "assimilated products" or "reanalyses"? And what do you mean by "variations in models"? Model uncertainties?

P6. L25. Are your results affected by the "ozonesonde problem" identified in Stauffer et al. (2020)?

P8. L21, L24, L26. Model data → reanalysis data, model profile → reanalysis profile, model → reanalysis.

P9 L9-16, See my general comment #2.

P9 L15-16. I don't understand this sentence. Could you rephrase it, please?

P9 L21 The symbol Δ. (the Laplace operator followed by a dot) should be replaced by ∇· (divergence)

P10 L8. Since you previously said that subscripts denote derivatives you can't use $M_y$ and $M_z$ as these are not the derivatives of M. I can't remember off-hand the notation in Andrews et al. but I suggest $M_{(y)}$ and $M_{(z)}$. I can't exclude the possibility that some authors use $M_y$ and $M_z$ but it strikes me as unnecessary abuse of notation.

P10 L15-16. The GEOS model used in the MERRA-2 data assimilation system does include chemical ozone production (albeit simplified). Do you simply mean to say that P and L is neglected in your analysis (Fig 11 and 12) but that doesn't lead to significant non-closure of the budget as evidenced in Fig. 11? Please, clarify.

P10 L20. "assimilated models" → data assimilation systems

P10 L20-24. I know what you mean but the way it's written this is somewhat contradictory. You say that the ground-based observations are too sparse, and then, in the next sentence you say that it's a "dense network".

P12 L2-3. It's not designed to do that. The MERRA-2 DAS does not include bromine chemistry.

P12 L11. How is significance evaluated? Also, note that lack of statistical significance does not necessarily mean that a result is not useful.

P13 L13. Larger percentage values may also result from low mean ozone concentrations in the denominator.

P13 L21. It's not clear to me what this means. Do you simply mean 75% of ozone molecules?

P14 L10. Why were those values chosen? Because they are representative of the vortex edge? Also, I think "105 K…" should be "$10^{-5}$ K…"; please use proper notation for the exponents. Note that 1 Potential Vorticity Unit = $10^{-6}$ $m^2$ $s^{-1}$ K $kg^{-1}$.

P14 L11. Approximately what altitude does 850 K correspond to?

P16 L4. Causality cannot be inferred from these maps alone. I suggested changing "caused" to "accompanied by" or something like that. You can explain that a causal mechanism will be established later in the paper.

P16 L3-11. Are the "elongated" and "displacement" events related to the more familiar characterization of SSWs as "displacement" and "split" or wave-1 vs. wave-2 events?

P16 L12-20 and Figure 6. This is really nice but I would like to see more detail. What latitude is the PV averaged over? What about the TCO? Is it evaluated over some region, latitude band or location? Can you provide a correlation coefficient and maybe draw a regression line? Let me say preemptively that I realize that the sample is small so technically there may be an issue with low statistical significance, but I wouldn't overestimate the importance of the latter, rather arbitrary notion.

P16 L24. Are these area-weighted averages?

P18 L2-4. I assume the "due to the effects…" bit refers to the 2006 situation. The wording suggests a causal link between the minor warming and the exceptional trajectory of the Greenland vs. zonal TCO in 2006 but no detailed analysis is presented to support that link, at least not until Section 6. Additionally, it looks to me like the increase is similar between the Greenland sector and the zonal mean also in 2013.

P18 L15. Is the zonal average taken between 60N and 80N as before? Are these area-weighted averages? Are the anomalies calculated with respect to the climatology?

P18 L25-27. "…because of". Again, causality is stated but not demonstrated until later.

P19 L15. I don't think that it's the vertical advection that leads to the poleward transport of PV. Isn't it rather due to planetary wave breaking and resulting mixing of low-PV low latitude air into the high latitudes?

P20 L20 The notation $M_y/dy$ is incorrect. If anything it should be $\partial M_{(y)}/\partial y$. Same for the vertical derivative. More troubling is the use of "y" (which was never defined, I believe; the equations so far were in terms of $\varphi$). These two quantities should be the terms of the del operator acting on **M**

in the spherical coordinates, i.e. one would expect something like $\frac{1}{a\cos\varphi}\frac{\partial}{\partial\varphi}(M_{(y)}\cos\varphi)$. Please make sure that the calculation is done correctly and that the notation is also correct.

P21 L8. Is this connection with "elongated vortex" something that you could explain, or at least provide a viable hypothesis for?

P22 L5. "*errors in the vertical derivatives over high latitude can be large as cos($\varphi$) gets small*)" Why? I would think that it's the horizontal derivative that would be sensitive to that, but it's very likely that I'm missing something obvious.

P22 L24. Again, it's not clear to me how significance is calculated.

P24 L23-25. As this (true) statement is not substantiated by the results of this study, it would be good to cite something here to support it.

P25 L8. Even adding just the 2021 SSW would help!

**Technical corrections**

P3. L1 Please double check the grammar. "SSWs" is plural

P3. L18-19. Grammar. "one of the strongest" → "some of the strongest"

P10 L5. The "a" (earth's radius) should be italicized.

P10 L10-11. The equation numbering changed from 1, 2,.. to 2.4, 2.5. Please, fix it.

P9 L22. "in this case ozone mixing ratio *tendency*"

Kris Wargan

**References**

Albers, J. R., Perlwitz, J., Butler, A. H., Birner, T., Kiladis, G. N., Lawrence, Z. D., … Dias, J. (2018). Mechanisms governing interannual variability of stratosphere-to-troposphere ozone transport. *Journal of Geophysical Research: Atmospheres*, *123*, 234–260. https://doi.org/10.1002/2017JD026890

Baldwin, M. P., Ayarzagüena, B., Birner, T., Butchart, N., Butler, A. H., Charlton-Perez, A. J., et al. (2021). Sudden stratospheric warmings. *Reviews of Geophysics*, *59*, e2020RG000708. https://doi.org/10. 1029/2020RG000708

Brasseur, G. P*., and* S. Solomon *(*2005*):* Aeronomy of the Middle Atmosphere*, 3rd ed., Springer,* New York*.*

Jaeglé, L., Wood, R., & Wargan, K. (2017). Multiyear composite view of ozone enhancements and stratosphere-to-troposphere transport in dry intrusions of northern hemisphere extratropical cyclones. Journal of Geophysical Research: Atmospheres, 122. https://doi.org/10.1002/2017JD027656

Orbe, C., K. Wargan, S. Pawson, and L.D. Oman, 2020: Mechanisms linked to recent ozone decreases in the Northern Hemisphere lower stratosphere. *J. Geophys. Res. Atmos.*, **125**, no. 9, e2019JD031631, doi:10.1029/2019JD031631.

Stauffer, R. M.,  Thompson, A. M.,  Kollonige, D. E.,  Witte, J. C.,  Tarasick, D. W.,  Davies, J., et al. (2020).  A post-2013 dropoff in total ozone at a third of global ozonesonde stations: Electrochemical concentration cell instrument artifacts? *Geophysical Research Letters*,  47, e2019GL086791. https://doi.org/10.1029/2019GL086791

Strahan, S.E., A.R. Douglass, and S.D. Steenrod, Chemical and dynamical impacts of stratospheric sudden warmings on Arctic ozone variability, *J. Geophys. Res. Atmos.*, *121*, 11,836–11,851, doi:10.1002/2016JD025128, 2016.

Tegtmeier, S.,  M. Rex, I. Wohltmann, and  K. Krüger, Relative importance of dynamical and chemical contributions to Arctic wintertime ozone, *Geophys. Res. Lett.*, *35*, L17801, doi:10.1029/2008GL034250, 2008.

---

## Referee Comment (RC2)

Review of

Analyzing ozone variations and uncertainties at high latitudes during Sudden Stratospheric Warming events using MERRA-2

submitted to ACP by Bahramvash Shams *et al.* (doi: 10.5194/acp-2021-646, 2021)

S. Chabrillat, BIRA-IASB, October 2021

**General Comments**

This study provides an interesting overview of the 6 SSW events which happened over the Arctic since 2004. Its main strength is the consistent usage of a leading reanalysis (MERRA-2) to compare the dynamical variabilities of these events and their relationships with the corresponding distributions of ozone. All analyzed fields (temperature, winds, ozone) come from the same reanalysis system, for all 6 events as well as for the climatology extracted from years with no SSW. This provides good confidence in the methodological consistency and in the validity of comparisons across the different years. This contribution to the field is sufficiently substantial to warrant publication in *ACP* after some revisions as outlined below.

These revisions may be considered minor because they probably do not require any new calculation (yet, see major comments 2 and 4). A general revision of the text is certainly necessary to address the first major comment.

**Major Comments**

1. The text should be improved w.r.t. consideration of related work and appropriate references. Citations do not seem very well used: it is difficult to see the links between specific results and specific references because these are always provided in groups. Not being an expert on SSWs, I often wondered what results are new and what results have already been published (e.g. for specific years or using less consistent datasets).
   Here are a few ideas and missing references to remedy this shortcoming:
   - How does MERRA-2 relate with other available reanalyses? ACP has a whole special issue about the SPARC Reanalysis Intercomparison Project (S-RIP) explaining that several similar reanalyses are available (Fujiwara et al., ACP, 2017) with different performances w.r.t. ozone (Davis et al., ACP, 2017) and providing, on a topic related to SSW, an assessment of ozone mini-hole representation in reanalyses over the Northern Hemisphere (Millán and Manney, 2017). The CAMS reanalysis (Inness et al., ACP, 2019) assimilated very similar ozone data as MERRA-2 and has been extensively validated (Wagner et al., doi:10.1525/elementa.2020.00171, 2021).
   - An extensive review about SSWs was published 8 months before the submission of this manuscript (Baldwin et al., 2021). Yet it is cited only to give the general definition of these events. This is a pity, because if would have be easy to contrast original results with results that are already discussed in this review.

- The manuscript nicely highlights the differences between elongated and displaced polar vortices prior to SSWs in the Northern Hemisphere. Has this distinction already been discussed w.r.t. ozone distribution in the Arctic stratosphere?
- The manuscript also highlights "the key role of vertical advection on mid-stratospheric ozone during the SSWs". Doesn't vertical advection also play a key role on mid-stratospheric ozone at other times and in other regions? What references have discussed this question?

2. P.5, line 21 that "...temperature, the northward wind (v), vertical pressure velocity ($\omega$), potential temperature ($\theta$, calculated from temperature and pressure), and potential vorticity (PV) are extracted from the pressure-level MERRA-2 dataset."
The pressure-level dataset has a coarser vertical resolution than the model-level (i.e. sigma-pressure) dataset. Since the TEM analysis (Figs 11-12) involves vertical derivatives, it should be performed on dynamical variables which are retrieved on model levels. Is it the case?

3. Figures 3 and 4, and related discussion (especially p.13, lines 12-17): How different are the corresponding diagnostics for years with no SSWs? Maybe one would obtain exactly the same biases and standard deviations of differences?
On these figures and throughout the text: the usual terminology is not "difference ratios" but "relative differences" or "normalized mean biases" and "standard deviations of the differences". See e.g. Lefever et al. (2015, doi:10.5194/acp-15-2269-2015).

4. P.19, lines 4-5: "The positive temperature anomalies in mid stratospheric layers start a few weeks before the SSWs in the 4 cases of a displaced vortex (2006, 2008, 2013, and 2019)."
But from the definition of the SSW (p.3 line 1) one of the two conditions to identify a SSW is an abrupt and intense increase of stratospheric temperature. Yet Figure 9 shows that the increase in temperature was not abrupt on these 4 years (those with displaced polar vortices). So one wonders how your algorithm for SSW identification could identify 2006/01/21, 2008/02/22, 2013/01/06 and 2019/01/02 ? I am also confused by the next sentence:
"On the other hand, the intrusion of the positive temperature anomalies to mid stratospheric layers is almost coincident with SSWs in the 2 elongated vortex cases."
But seeing the definition of SSWs, shouldn't this be a feature of all SSWs?
I think that this should be clarified not only in the author's response but also in the revised manuscript.

**Tables and figures**

- Table 1: What is the "full PCO" in "%full PCO uncertainties"? Does it mean "Partial Column of Ozone"? But for what pressure range? Or maybe that is the TCO?
  Do these uncertainties (two rightmost columns) come from Bognar et al. (2019) or are they a new result of this paper?

- Figure 1: Add a box for the Greenland sector; stretch the color scale towards the reds in order to increase the contrasts

- Figure 2: Add a sentence to the caption, e.g. "***The vertical red lines highlight the dates of the 2008 and 2009 SSWs***".

- Figures 3 and 4: re-formulate the captions to obtain similar captions while avoiding the words "difference ratios". Consider: "***Normalized mean biases and standard deviations of MERRA-2 with respect to ozonesondes/FTIR observations***".

- Figure 5: totally unreadable, even at maximum zoom on a large screen! You must change the layout of the figure to decrease the margins around each map and increase its relative area and increase the resolution of the bitmap or (better) save these maps as vector-oriented graphics (PDF).
  Please clarify the caption: "***Mean values*** of the TCO anomalies..."

- Figure 8: Please remind in the caption, for the casual reader: "...averaged over the  ***latitude band 60°N-80°N*** and ***over the*** Greenland sector ***(60°N-80°N, 10°W-70°W)***."

- Figure 10: How different is the time evolution of w* on a year with no SSW? One expects smaller and less perturbed values, but by how much? Consider adding the same figure but from the climatology of years with no SSW

- Figure 11: these plots should not show results below 15 km because these results cannot be discussed since
  "Considering the larger uncertainties of ozone estimation in MERRA-2 below 15 km, and the possibility of larger uncertainties in dynamic parameter estimations, this study does not analyze the impact of the dynamics on ozone in the lower stratosphere." (p. 21, lines 17-20).

**Minor Comments and typos**

- P.1, line 17: clarify "***During SSWs***, changes in..."

- P.3, lines 10-11: "... many other factors such as lower stratosphere conditions, the geometry of the polar vortex, the gradient of potential vorticity (PV) at the edge of the polar vortex, and synoptic systems at lower altitudes (Tripathi et al. 2015, de la Cámara et al., 2019; Lawrence and Manney, 2020). Changes in momentum deposition associated with these processes leads to..." These are not "processes". Maybe "conditions" or "dynamical states"?

- P.4, lines 15-16: improve transition with next paragraph, e.g.
  "This study investigates ***dynamical variability and*** ozone variations ***above the Arctic*** (between 60***°N*** and 80***°***N) ***both*** in the zonal average ***and above a specific*** region, during six SSWs using the MERRA-2 dataset."

- P.4, line 20: this is the first occurence of "Greenland sector" so you should move here your definition of this region (currently on p.16) and also draw the corresponding box on Fig. 1.

- P.5, lines 7-10 and 22-23: this attempt to define the MERRA-2 reanalysis and its assimilation system is not correct. Reanalysis systems use only one model, here GEOS5; a well-designed reanalysis does not have any variations in models nor in methods of analysis – only in assimilated datasets of observations. See e.g. Fujiwara et al. (ACP, 2017) for a general yet correct description of reanalysis systems such as MERRA-2.

- P.6, lines 5-6:
  "...atmospheric dynamics, displaced/split polar vortex, and hemispherically asymmetric conditions during SSWs may cause unusual nonlinearity in ozone flux/transport terms."
  What do you mean by "unusual nonlinearity"?

- P.7, lines 16-18: "Having the ability to resolve the fine structure of solar radiation spectra allows the retrieval of a variety of trace gases using the NDACC solar FTIR. However..."
  This sentence is irrelevant for this paper – consider removal.

- P.7, line 21-22: this citation of Bognar et al. (2019) does not seem to belong here as they validate satellite instruments – not ground-based instruments?

- P.8, line 1: "More details of ***on*** the ozone retrievals at Eureka"

- P.8, lines 16-17: "However, the vertical resolution of the remote sensing retrieval is often not similar to the model grid points...". Consider instead:
  "***Since*** the the vertical resolution of the remote sensing retrieval is ***much coarser than the vertical resolution of the model***..."

- P.8, eq. (1): Do $x_s$, $x_h$ and $x_a$ represent vertical profiles of ozone ***mixing ratios***?  Please clarify.

- P.8, line 26: "The smoothing method effectively linearizes the ozone from the model..."
  I do not understand. How can ozone, or even its mixing ratio, be "linearized" ?

- P.10, lines 15-16: "the lack of net chemical production in the assimilation model should not dramatically impact our conclusions."
  Could heterogeneous chemistry losses happen in Polar Stratospheric Clouds prior to the SSWs?

- P.11, lines 15-20: this is a pure repetition of details already given in the caption of the figure. I recommend to keep this in the figure caption and to remove it here.

- P.11, lines 23-24, consider:
  "the 5km-10km **layer** includes **the** upper troposphere **and the** lower**most** stratosphere (UTLS), while the 10-30km layer includes the lower **middle** stratosphere**.**"

- P.12, line 2: "MERRA-2 is  **appears** unable to retrieve..."

- P.13, line 1: "...contain the most column ozone...".
  Consider instead: "***...contribute most to the total ozone column...***"

- P.13, line 24-25: re-write the sentence. Consider e.g. :
  "...our primary analysis is focused on the mid-stratospheric layers **which contribute most to the TCO and**  where the measurements are most reliable ."

- P.14, lines 8-12: please split this sentence in several parts to make it clearer. Specifically, I understood only later on that you plot and discuss the ***mean values over the 15 days prior to the SSW and over the 15 days that follow***. Please state this clearly already here and also in the caption of Fig.5.

- P.14, lines 22-23: do you mean
  "The easterly winds lasted 15 days  **atfer** the major warming on 22 February." ?

- P.16, line 8: what do you mean by "semi-symmetrical shape"?

- P.16, lines 21-23: if we zoom on Fig.15 to the maximum possible and on a large screen, we can see (despite the insufficient resolution of the figure) that this characterization does not hold for the 2019 SSW. See also major comment on unreadability of Fig. 5.

- P.17, line 3: "the Greenland sector exhibits a very strong isolated stratospheric air circulation during wintertime..."  Consider instead: "***The air masses above the Greenland sector are more strongly isolated than at other Arctic longitudes***".

- P.17, line 7: "...and the Greenland sector  **from** 40 days before to 60 days after each SSW..."

- P.17, line 14: "The climatological polar vortex position is located over the Greenland Sector (Figure 1)". I do not agree: the center of the climatological vortex, as shown on Figure 1, is above Ny Alesund which is sligthly outside (East) of the Greenland sector as defined here.

- P.17, lines 18-19: "... over the Greenland sector  **with a** larger drop  **of** EPV **in this region than in**  the zonal **mean**."

- P.18, lines 1-2: "In all SSWs the  **relative increase of the** TCO is higher over the Greenland sector compared to the zonal average with the exception of 2006..."
  Is this significant? It seems to me that the values above Greenland are barely larger than at other longitudes.

- P.18, lines 5-12: is this paragraph useful or interesting? Consider deletion.

- P.18, lines 22-23: split the sentence: "...for both the zonal averaged and the Greenland sector*.* ***As*** expected the structures of ozone anomalies are smoother in the zonal average compared to the Greenland sector ."

- P.18, lines 24-25: this sentence is very unclear. Are you still comparing the Greenland sector with the zonal average? Please re-write.

- P.18, line 26: "The shortest impact on TCO  ***happened in*** 2008..."

- P.19, lines 15-17: "...which leads to ***poleward*** advection of low EPV air parcels . The conservation of EPV causes anticyclonic circulation, which gradually drives easterly ***the*** zonal mean zonal winds, and leads to ***the*** displacement or splitting of the polar vortex."

- P.20, line 1, consider: "Occurrences of minor SSWs  ***can be seen through*** the early appearance"

- P.20, line 8, consider: "The suppressed wave activity  ***leads to*** the recovery of..."

- P.22, line 7: Section 7 provides a summary, with only the last paragraph providing a conclusion. Hence the title of this section should be "**Summary and conclusion**".

- P.22, line 13: "***The*** MERRA-2 reanalysis..."

- P.22, lines 24-25, consider: "From 5km to 10km, a  negative mean bias exists in all sites (-8% to 15%) ***but it is not significant due to the larger standard deviation***."

- P.23, line 1: please describe "G-5km (<20%)" properly, with words

- P.23, line 8: "These results emphasize the high quality of MERRA-2 ***at least after 2004, the year when MLS data became available.***" This is important!

- P.23, lines 9-10: "Higher uncertainties in ***the*** UTLS are also expected because MLS has ***a dominant contribution in the MERRA-2 reanalysis and a*** lower sensitivity at lower altitudes "

- P.23, line 22: "The TCO increase rates and the magnitude of change in EPV after these cases are large and the intrusion of positive temperature anomalies to the mid stratosphere is coincident with ***these*** SSWs date."

- P.24, lines 3-5, consider: "A strong *cor*relation is observed between the magnitude of change in the averaged EPV  ***around*** the SSW, and the magnitude of TCO change for the same period for all six studied SSWs."

- P.24, lines 6-15: "The Greenland sector is one of the critical regions that is impacted by negative TCO *anomalies* before the elongated polar vortex in 2009 and 2018; positive TCO *anomalies* occurs before displaced SSWs. To identify the similarities and differences of zonal versus the regional impact of SSWs on ozone, the analyses are applied over the Greenland sector as well as the zonal average. The general structure of the vertical ozone anomaly over the Greenland sector is similar to the zonal *structure*. However, as expected the ozone anomaly over the zonal average is smoother than *over* the Greenland sector which results in a more magnified TCO increase over Greenland. The increased rate over the Greenland sector is between 15% in 2006 to 38% in 2018, while the zonal average ranges between 8% in 2008 to 29% in 2009. Moreover, *the* TCO exhibits a faster recovery to the climatology values over this region compared to the zonal average."

- P.24, line 26: not understandable – please re-write:
  "The faster recovery of zonal temperature and ozone at middle stratosphere within 30 days is recorded for 2008 with the shortest duration easterly zonal mean zonal winds."

- P.25, line 3-6: "In conclusion, the MERRA-2 dataset is shown to capture *the* ozone variability in the middle stratosphere and provides dynamical information to investigate the impact of SSWs. The impact of SSWs on ozone and the role of vertical advection is shown to be more intense in 2009 and 2018 with an elongated polar vortex compared to *the* displaced  *vortices* in 2006, 2008, 2013, and 2018."

- P.25, last sentence: "the dramatic ozone increases over high latitudes during SSWs points to consequences of these events on the global earth system and possible environmental/ecosystem changes that could be investigated in future studies."
  This is a quite vague statement and I am skeptical as the timescales for environmental/ecosystem changes are much longer than those due to SSW perturbations. Maybe it is possible to conclude instead with the changes in SSW occurences that are expected from climate change?

---

## Author Comment (AC1)

Authors response:

We are thankful to both reviewers for their thorough comments. We have added a few paragraphs to the introduction to better clarify previous studies that are relevant to this manuscript and to elucidate remaining research questions that this study focuses on. A few graphs have been updated to have higher resolution and to include statistics, as suggested by the reviewers. We have used more clear language to highlight the novelty and main findings of this study. We applied most of the reviewer's suggestions and provide justification for those that we did not include. We believe that the quality of this manuscript has been greatly improved by these reviews.

Our detailed responses are given below, as well as comments on the full manuscript.

To facilitate reading of our responses and tracking changes in the full manuscript, we have labeled each of the reviewer's comments using the following labels:

List of labels:

reviewer 1 (Kris Wargan) :
        General comments (GC+#)
        Specific comments (Spc+#)
        Technical corrections (Tc+#)
reviewer 2 (Simon Chabrillat):
        General comments (GCC+#)
        Tables and figures comments (TF+#)
        Minor Comments and typos (MC+#)

We also used different fonts, sizes and colors: Reviewer comments are in black in Calibri font, our written responses are in blue in Calibri font, and the updated text in the manuscript is in cyan in Time New Roman font using a small font size.
* * *
**Review of the manuscript "*Analyzing ozone variations and uncertainties at high latitudes during Sudden Stratospheric Warming events using MERRA-2*" by Shima Bahramvash Shams et al.**

This study examines the evolution of polar ozone during six recent major sudden stratospheric warming events (SSW) using the MERRA-2 reanalysis. The analysis is preceded by evaluation of the MERRA-2 ozone using independent observations and focusing on the region and periods of interest, which provides a nice complement to previous validation studies that looked mainly at the global picture. The authors find that the impacts of SSWs on ozone were largest in 2009 and 2018 ("elongated vortex" cases) compared to the other events ("displaced vortex") and identify

vertical advection as the key mechanism responsible for the formation of these positive ozone anomalies during all the SSWs analyzed in this study.

This paper is interesting and certainly suitable for publication in ACP subject to some minor revisions as delineated below. It's nice to see a detailed study of a number of recent SSWs in one place along with an ozone budget analysis for each of them. It certainly adds to our understanding of the role of transport during these events. It's also great to see another paper that demonstrates the value of reanalysis ozone data. It is only recently that our community began to take advantage of the global coverage, high resolution and dynamical consistency that these products offer. The additional validation of MERRA-2 ozone, focused on the region of interest is especially valuable. While I ended up having a rather large number of comments and suggestions, none of them are serious objections, and I believe all of them can be easily addressed. The most important ones concern the need to highlight the novel aspects of this work in the context of other similar studies (some of them not cited) (general comment #1), the choice of these particular six events and a possibility of adding 2021 to the list (general comment #2), and terminology (#3). I hope my comments will be useful.

**General comments**

GC1.        Can you place your results in the context of previous papers discussing the role of dynamical ozone resupply, e.g., Tegtmeier et al., 2008; Strahan et al. (2016)? It would help if you clearly delineated the novel aspects of your study against the backdrop of the existing literature of the subject.

Response:

We have added multiple paragraphs in the introduction to provide better connect to previous studies, which now focus our manuscript on key research questions. Also, some sentences are added to the abstract to provide better focus on the key findings: the average shape of polar vortex before SSW, EPV changes to the geographical extent, timing, and magnitude of ozone changes, and, finally, the magnified vertical advection in the elongated vortex shape.

[revised manuscript text omitted]

As Tegtmeier et al., 2008 and Strahan et al. 2016 both focus on springtime impact on ozone when chemistry is significant modulator of Arctic ozone in the middle stratospheric, we added these articles in our methodology section and discuss why chemistry is not significant modulator of ozone changes during winter over polar night.

"The contribution of dynamical and chemical drivers of ozone anomalies varies throughout the year. During springtime, both dynamical resupply and chemical depletion strongly modulate ozone changes. Assuming an isolated polar vortex and neglecting isentropic mixing, a previous study showed a similar magnitude of influence from chemical ozone depletion processes and dynamical ozone supply during the springtime (Tegtmeier et al. 2008). However, Strahan et al. (2016) used a chemistry and transport model to show that dynamical processing affects ozone changes by a factor of two more than chemical processing during March. However, chemical processes are not significant drivers of ozone changes in the middle stratosphere from November to February in the Arctic because of the polar night (de la Cámara et al. 2018b). Moreover, it has been shown that during years with SSWs, Arctic ozone depletion is significantly diminished (Strahan et al. 2016). However, if prior to or during the SSWs, the polar vortex moves outside of the region of the polar night (to lower latitudes), ozone depletion will occur as shown in the 2013 SSW by Manney et al. (2015). By limiting our analysis to latitudes between 60ºN to 80ºN, this impact is minimized in our analysis."

GC2. Butler et al. (2017) as well as the SSW Compendium (https://csl.noaa.gov/groups/csl8/sswcompendium/majorevents.html) list several more SSWs than those discussed in the paper: 2007, 2008, and 2010. I assume there's a good reason for the selection discussed in this study to be what it is, but it needs to be explained, especially since the definition of SSW used here is the same as that applied in the Compendium. In addition, ideally, I would like to see 2021 added to the analysis. Seven events are better than six. Note that it would likely be the first paper that talks about ozone during the 2021 SSW.

Response: In this study, we focused on persistent SSWs with at least 16 days. As shown by Lee and Butler (2019), the average duration of SSWs is 12-13 days. The SSW in 2008 has been part of our analysis as the least strong SSWs with 16 days of easterly days. The SSW in 2007 exhibit only 4 days of easterly zonal mean zonal winds. The easterly zonal mean winds last for 9 days in Feb 2010. Moreover, the increase in stratospheric temperature in 2010 that occurs in mid-January is not synchronized with the wind reversal. Thus, in this study, we focused on six persistent SSWs between 2004 and 2020. Even 2008 shows a very small impact in our analysis, thus, to not overwhelm our plots and analysis, we focus on these six events that are also among the top ten strongest events since 1979. However, to show that our conclusion about the EPV change and ozone changes is valid for weaker events, these two events are added to the regression plot (Fig 6). This plot also emphasis that these two events are the same level of influence as SSW in 2008 with regard to EPV change and ozone changes.

Addition text to sec 3:

"This paper focuses on six persistent mid-winter (December-February) major warmings in this period that exhibited persistent easterly zonal mean zonal winds with a duration of at least 16 days (Table II). Table II includes the duration, magnitude of the easterly zonal wind, and the duration of polar vortex recovery for each SSW; all information is derived from MERRA-2 data. It should be noted that the duration of the easterly wind shown in Table II is not necessarily consecutive. Two major SSWs during the 2004-2020 time period are not included in the main results of our study because they did not meet the persistence criteria. The major SSW in 2007 exhibits only 4 days of easterly zonal mean zonal winds, while the major SSW in Feb 2010 exhibits only 9 days. However, SSWs in 2007 and 2010 are included in the regression analysis for Figure 6 for more robust statistics which also shows that they had some of the lowest impact on ozone."

Additional text to sec 5:

To increase the robustness of the regression analysis, SSWs in 2007 and 2010 are also included here (Fig. 6).

We believe that by using the suggestions and feedback by both reviewers we have clarified the novelty of the work using six persistent events between 2004 to 2020. We briefly looked into the SSW in 2021 and considering its duration of 15 days of easterly winds with winds only reaching -10 m/s, it was a weak event compared to our original set of events. Thus, we decided to keep the scope of this study between 2004 to 2020.

GC3. There is a terminological confusion regarding the use of the word "model" where you really mean either data assimilation system, assimilated data, or reanalysis. I tried to catch those instances in my specific comments below. I think it's important to remember that a reanalysis is **not** a model simulation. Rather, it is fundamentally a data-driven product. Calling a reanalysis a "model" is equivalent to calling a retrieved satellite data set "a radiative transfer model". The role of the general circulation model in a data assimilation system is to propagate information from observations in time and space over a period of several hours. That's all it does. Things do get a bit muddled in places/periods devoid of observations, but your study looks at regions well constrained by data, so that's not an issue here.

Response: All occurrences are changed to data and system as commented in the text.

GC4.        You use the terms lower and middle stratosphere somewhat loosely. Please define them somewhere in the methods section.

Response: In methods section, we added a sentence to emphasis the middle stratospheric layers that will be the focus of the further analysis:

> "In further sections, analysis will focus on middle stratospheric layers between 15 and 30 km."

For lower stratosphere, in the section 4, we defined the UTLS in the layer description (p11, l27):

> "5 to 10 km includes the upper troposphere-lower stratosphere (UTLS)"

GC5.        You choose to exclude the lower stratosphere from this analysis citing the high uncertainty of MERRA-2 ozone there. While it's completely OK to make that choice I have a problem with the stated motivation. It's certainly true that the uncertainties get larger closer to the tropopause, but I don't think one can say, based on the validation that you did or the results in Wargan et al (2017), that there's no useful information down there, i.e. that the true variability is completely obscured by uncertainty. The ozonesonde comparisons in Fig. 3 suggest difference standard deviations of about 25%. Below I plot the time series of MERRA-2 ozone at Eureka at 150 hPa along with 25% and 50% envelopes. Clearly, the dynamical variability around the 2009 SSW is still very much discernible in the sense that the magnitude of the large jumps exceeds the uncertainty. Additionally, note that Albers et al. (2018) found useful ozone information in MERRA-2 in the lowermost stratosphere and Knowland et al. (2017) as well as Jaeglé et al. (2017) successfully studied stratospheric intrusions into the troposphere using MERRA-2 ozone. Again, I agree that any results in the LS would have larger uncertainty, but I can't agree that they would be worthless, and the present wording in the paper seems to imply that. I'm not suggesting extending the analysis. It's just a matter of phrasing things in a more nuanced way.

Response: We agree with the reviewer comment. The text is updated to reflect this comment:

> "The larger uncertainties below 10 km during the five months impacted by SSWs are consistent with larger uncertainties in MERRA-2 in these layers year-round, as seen in previous studies (Gelaro et al. 2017; Wargan et al. 2017). However, large fluctuations within the lower atmosphere ozone are still discernible from MERRA-2 data (Knowland et al. 2017; Jaeglé et al. (2017); Albers et al, 2018).
>
> ..
>
> Because more than 80% of ozone molecules exist in the middle stratosphere (10 to 30 km), the total column uncertainty is dominated by uncertainties in mid-stratospheric layers. In the following section, we discuss ozone variability in the total column and the vertical profile up to 60 km, while our primary analysis is focused on ozone and dynamical processes the mid-stratospheric layers, which contribute most to the TCO and where the measurements are most reliable. "

GC6.        In several places the language of the paper implies (or even outright) states causal relationships where causality is not immediately demonstrated. I provide some examples in my specific comments. There are sentences like "*the vortex displacement toward the southeast (Europe) prior to the 4 major SSW as seen in 2006, 2008, 2013, and 2019 (hereafter the displaced vortex SSWs) caused an early positive ozone anomaly*", "*an elongated polar vortex is observed before the SSWs which caused a dominant negative ozone anomaly*", "*SSWs and their impact on ozone*" (the title of a section). I get it that Figures 11 and 12 and the discussion in Section 6 do provide evidence for a causal mechanism, but the reader doesn't know that in advance. I suggest rephrasing these causal statements as something more "relational" (Fig 6 and the wording on P24 L3-5 are very good) and explaining that a dynamical mechanism will be elucidated in Section 6. You can then state your results in causal terms in the conclusions as you do. It's really about streamlining the argument.

Response: We agree with the reviewer comment and fixed the language. Also, the revised manuscript places more emphasis on Figure 6, plus the new categorization of the polar vortex before the event, to identify the impact on ozone. As discussed in the response to GC1 and some parts of the discussion.

**Specific comments**

**Spc1:** P2. L19-22. I believe that this "ozone gets made in the tropics and transported to the high latitudes" description is simplified to the point of being incorrect (albeit quite commonly used). Please, take a look at the discussion around Fig. 5.11 in Bresseur & Solomon 2005 (in my edition it starts on page 286 with "It is sometimes stated that ozone is produced where its mixing ratio maximizes…"). This does not affect anything in your paper of course.

Commented [1]:
Spc23

Response: In that sentence we referred to ozone accumulation during wintertime over high latitudes that is modulated by BDC (rather than the absolute source of ozone in the Arctic). We rephrased the sentence to avoid any confusion for readers.

> "High latitude ozone accumulation during winter and peak values in the spring are largely controlled by BDC transport of ozone-rich, tropical stratospheric air."

**Spc2:** P2. L24-25. Would it make sense to cite the recent review paper on SSWs by Baldwin et al., 2021?

Response: It is added to the text.

**Spc3:** P4. L3-4. "numerical and assimilation models". I struggle with this terminology. I understand that by "numerical models" you mean general circulation models or numerical weather prediction models. I can live with the short-hand version "numerical models". But I think the word model should not be applied to data assimilation systems, which comprise a model component and a statistical analysis scheme. I suggest "numerical models and data assimilation systems". See my general comment #3.

Response: The text is updated:

> "The complexity of altered dynamics of SSWs might introduce extra uncertainties into the numerical models and data assimilation systems."

**Spc4:** P4. L15-16. At first read this sentence confused me: why focus on zonal structures? SSWs are very non-zonal! But you're actually doing much more than that: the Greenland sector, polar averages, maps. I suggest rephrasing it.

Response: That paragraph is completely revised as discussed in respond to GC1.

Related to this comment, we also In section 7, page 24:

> "The variability in impact of SSWs on high latitude ozone is analyzed, two different patterns are found, and the possible dynamical mechanisms involved, are studied."

**Spc5:** P4. L17. "assimilation model". See above. Why not just say reanalysis?

Response: It is updated to reanalysis data.

**Spc6:** P5. L7-9. The description of MERRA-2 needs some rewriting. It's not clear to me what it means that there's a variety of models incorporated in MERRA-2. Do you mean GCM, land model, parameterized chemistry, etc.? There aren't multiple "general circulation model**s**" in it

as the text implies. There is one. Also, I don't know what you mean by "extended reanalysis". No other reanalysis is incorporated in MERRA-2. If you mean something like OSTIA (the SST data set that provides boundary conditions for the GCM), it's not a reanalysis, at least not in the same sense as MERRA-2.

Response: The text has been updated to apply the reviewer's comment:

> "A variety of data sets are incorporated into a general circulation model to create 3-dimensional MERRA-2 ozone datasets with a time-frequency of 3 hours (Wargan et al., 2017; Gelaro et al., 2017)."

**Spc7:** P5. L16. I wouldn't say that it's been extensively used in trend studies. To my knowledge only Wargan et al. (2018) derived trends from (suitably corrected for discontinuities) MERRA-2 (there was a follow-on paper by Orbe et al. 2020, but MERRA-2 ozone is sort of tangential there). Also, please consider adding a very interesting study by Albers et al. (2018) to these citations.

Response: We removed the extensive from the sentence and added Albers et al. (2018) to the citation list.

**Spc8:** P5. L21. Please, provide a citation for the data collection used in this study. Information on how to cite the MERRA-2 pressure-level output is here: https://disc.gsfc.nasa.gov/datasets/M2I3NPASM_5.12.4/summary under "data citation".

model-level output: https://disc.gsfc.nasa.gov/datasets/M2I3NVASM_5.12.4/summary (I'm providing both in case I misunderstood which one you used; see the next comment).

Response: The data source is now cited (as shown in the next response)

**Spc9 :**P5. L21. Why use pressure-level output if much higher vertical resolution model-level output is available? But P8 L16 talks about using the model levels. So which collection is really used?

Response: To have the finest vertical resolution for the comparisons with observations, MERRA-2 ozone in the model level is used (GMAO, 2015a). To investigate dynamical mechanisms, using pressure-level data facilitate estimation of variables such as PV and potential temperature and consequently ozone in pressure level is used for associated analysis. We converted the pressure coordinate to geopotential height to estimate the derivatives of further analysis.

> "To have the finest possible vertical resolution for the comparisons with observations, MERRA-2 ozone at the model levels is used (GMAO, 2015a). Other dynamical variables such as temperature, and the northward and vertical wind velocities ($v$, $\omega$), are extracted from the pressure-level MERRA-2 dataset (GMAO, 2015b), which facilitates the calculation of variables such as potential vorticity (PV) and potential temperature ($\theta$)."

**Spc10:** P5. I think it should be mentioned that MERRA-2 assimilates MLS ozone down to 177 hPa between 2004 and 2015, then switches to version 4.2 retrievals going down to 215 hPa. This version (and vertical range switch) results in some differences between the pre- and post 2015 periods, whereby the latter has likely more accurate ozone, especially in the lower stratosphere.

Response: It is applied.

> "Total column ozone from the Solar Backscatter Ultraviolet Radiometer (SBUV) (1980 to 2004) and the Ozone Monitoring Instrument (OMI) (since 2004) and retrieved ozone profiles from SBUV (1980 to 2004) and the Microwave Limb Sounder (MLS) (since 2004, down to 177 hPa to 2015, down to 215 hPa after 2015) are used to estimate ozone in MERRA-2 (Gelaro et al., 2017)."

**Spc11:** P5. L22. "Assimilated/reanalysis models". Why not just say "assimilated products" or "reanalyses"? And what do you mean by "variations in models"? Model uncertainties?

Response. The sentence is updated.

> "In reanalysis products such as MERRA-2, methods of analysis, model uncertainties, and observations cause uncertainties in the products (Rienecker et al. 2011)."

**Spc12:** P6. L25. Are your results affected by the "ozonesonde problem" identified in Stauffer et al. (2020)?

Response: Only Eureka is among the affected stations as discussed by Stauffer et al. (2020), and the artifact is observed starting from early 2016. As shown in the appendix of the same study, starting in 2018, the differences are back to normal (Fig B of appendix of Stauffer et al. (2020)). None of the SSWs studied here are during the period of artifacts. Thus, our comparisons are not affected by this issue.

**Spc13:** P8. L21, L24, L26. Model data→reanalysis data, model profile→reanalysis profile, model→reanalysis.

Response: The text is updated.

**Spc14:** P9 L15-16. I don't understand this sentence. Could you rephrase it, please?

Response: It is updated.

> "This study also analyzes the impact of different dynamical transport mechanisms on ozone for each of the major SSWs."

**Spc15:** P9 L21 The symbol Δ. (the Laplace operator followed by a dot) should be replaced by $\nabla \cdot$ (divergence)

Response: It is updated.

**Spc16:** P10 L8. Since you previously said that subscripts denote derivatives you can't use $M_y$ and $M_z$ as these are not the derivatives of M. I can't remember off-hand the notation in Andrews et al. but I suggest $M_{(y)}$ and $M_{(z)}$. I can't exclude the possibility that some authors use $M_y$ and $M_z$ but it strikes me as unnecessary abuse of notation.

Response: It is updated to $M_{(y)}$ and $M_{(z)}$.

**Spc17:** P10 L15-16. The GEOS model used in the MERRA-2 data assimilation system does include chemical ozone production (albeit simplified). Do you simply mean to say that P and L is neglected in your analysis (Fig 11 and 12) but that doesn't lead to significant non-closure of the budget as evidenced in Fig. 11? Please, clarify.

Response: The text is updated:

"Thus, neglecting P and L below 30 km in further analysis, as chemical production and loss is not an output of reanalysis data, does not lead to significant non-closure in the presented analysis and does not impact our conclusions."

**Spc18:** P10 L20. "assimilated models"àdata assimilation systems

Response: The text is updated.

**Spc19:** P10 L20-24. I know what you mean but the way it's written this is somewhat contradictory. You say that the ground-based observations are too sparse, and then, in the next sentence you say that it's a "dense network".

Response: The text is updated:

"However, the use of ground-based observations to directly study the impact of SSWs is challenging because of the coarse time resolution of ozonesondes, limited clear-sky conditions and sunlight for FTIR measurements, and dealing with one profile per site/launch time for each sensor, and its subjectivity to the site location and time."

**Spc20:** P12 L2-3. It's not designed to do that. The MERRA-2 DAS does not include bromine chemistry.

Response: The text is updated.

"The extreme low ozone values near the surface are not represented in MERRA-2 as it does not include bromine chemistry."

**Spc21:** P12 L11. How is significance evaluated? Also, note that lack of statistical significance does not necessarily mean that a result is not useful.

Response: the text is updated.

> "From 5 to 10 km, a negative mean bias exists at all sites however they are accompanied by a larger the standard deviation."

**Spc22:** P13 L13. Larger percentage values may also result from low mean ozone concentrations in the denominator.

Response: We expect that the differences of low concentration should be smaller too. Even if we consider a detection limit for low values that is a source of uncertainty at lower levels.

**Spc23:** P13 L21. It's not clear to me what this means. Do you simply mean 75% of ozone molecules?

Response: The text is updated to ozone molecules.

**Spc24:** P14 L10. Why were those values chosen? Because they are representative of the vortex edge? Also, I think "105 K..." should be "$10^{-5}$ K..."; please use proper notation for the exponents. Note that 1 Potential Vorticity Unit = $10^{-6}$ m$^2$ s$^{-1}$ K kg$^{-1}$.

Response: We updated the range to fit the standard PV units of 600 and 800 ($10^{-6}$ K m$^2$ Kg$^{-1}$ s$^{-1}$). We used these two values of PV that show the extent of the polar vortex. Although these values are not the exact edge of the polar vortex, even using the maximum gradient of PV as polar vortex edge has its own uncertainties (as discussed in Serra et al. 2017). Because we are looking at 15-day averages, the fine structure of polar vortex locations are smoothed and fix contours provide a simple and easy way of estimating polar vortex general intensity and positioning.

**Spc25:** P14 L11. Approximately what altitude does 850 K correspond to?

Response: ~ 30km, and it is added to the text.

**Spc26:** P16 L4. Causality cannot be inferred from these maps alone. I suggested changing "caused" to "accompanied by" or something like that. You can explain that a causal mechanism will be established later in the paper.

Response: The text is updated and now uses "accompanied".

**Spc27:** P16 L3-11. Are the "elongated" and "displacement" events related to the more familiar characterization of SSWs as "displacement" and "split" or wave-1 vs. wave-2 events?

Response: One of the novelties of this study is showing ozone changes due to SSWs are connected to the averaged polar vortex shape before the SSW as we categorize them as elongated or displaced. These averaged polar vortex states are different, though often related to, split and displaced vortex breakdown conditions. As seen during the SSWs in 2018 and 2009, the polar vortex splits and the 15-day average polar vortex before that event is elongated. Other events such as 2013 and 2019 first displace, and then split. However here we considered the displacement as the 15-day average as displaced and not elongated.

Additional text to section 5, page 17:

> "The averaged polar vortex state we refer to in this study is different, though often related to, split and displaced vortex morphology discussed in previous literature (e.g., Charlton and Polvani 2007). As seen during the SSWs in 2018 and 2009, in which the polar vortex split, the 15-day average polar vortex before those events is elongated. Other events, such as those in 2013 and 2019, first displace and then split. However, here we consider them displaced SSWs if the 15-day average EPV prior to the event is displaced and not elongated. Previous studies focused on the connection of the type of polar vortex breakdown to its impact on the speed of trace gas transitions (Charlton & Polvani (2007); Manney et al. 2009b). This study investigates the modulation of the magnitude and extent of ozone changes, and the results show that the average EPV shape before the vortex breakdown is more influential than the final form of polar vortex breakdown."

Additional text to section 5, page 18:

> This result shows that the averaged polar vortex shape before the SSWs is connected to the EPV change and then dramatically influences the magnitude of ozone changes at high latitudes.

**Commented [BSS2]:** Spc28

**Spc28:** P16 L12-20 and Figure 6. This is really nice but I would like to see more detail. What latitude is the PV averaged over? What about the TCO? Is it evaluated over some region, latitude band or location? Can you provide a correlation coefficient and maybe draw a regression line? Let me say preemptively that I realize that the sample is small so technically there may be an issue with low statistical significance, but I wouldn't overestimate the importance of the latter, rather arbitrary notion.

Response: The plot is updated to have the regression line and R2. And the text is updated: A correlation between the magnitude of change in EPV and TCO is very strong (R2= 90%). We emphasize this finding more through out the paper and, in the introduction, and discussion.

> "To investigate the connection of polar vortex strength and TCO, the scatter plot of the zonally averaged (60ºN to 80ºN) EPV change at the potential temperature of 850 K versus the corresponding change in TCO (60ºN to 80ºN) is shown in Figure 6. All averages are area weighted, and the ratio of change for each variable is estimated as the average of 15 days after SSWs subtracted by the average of 15 days before the SSWs and divided by the average of 15 days before the SSWs. To increase the robustness of regression analysis, SSWs in 2007 and 2010 are also included here (Fig. 6). The correlation between the magnitude of change in EPV and TCO is very strong (R2= 90%). The elongated vortex SSWs (2009 and 2018) exhibit a higher magnitude of change in both EPV and TCO in this period. This result shows that the averaged polar vortex

shape before the SSWs is connected to the EPV change and then dramatically influences the magnitude of ozone changes at high latitudes."

Also, the updated text in the conclusion, page 25:

"A strong correlation of $R^2 = 0.90$ is observed between the magnitude of change in the averaged EPV around the SSW and the magnitude of TCO change for the same period for all six studied SSWs and including two less persistent SSWs in 2007 and 2010. The regression analysis also emphasized larger changes in both EPV and TCO during elongated SSWs."

page 26:

"This study shows that the averaged vortex shape before the SSWs is an important modulator of the magnitude and extent of ozone changes over high northern latitudes."

**Spc29:** P16 L24. Are these area-weighted averages?

Response: Yes. We added the area-weighted to the text:

"the variability of area-weighted ozone average over the Greenland sector (60ºN to 80ºN and 10ºW to 70 ºW) as well as the zonal average (60ºN to 80ºN) is analyzed to investigate the similarities and differences of the impacts of SSWs on zonal and regional high latitude ozone"

**Spc30:** P18 L2-4. I assume the "due to the effects..." bit refers to the 2006 situation. The wording suggests a causal link between the minor warming and the exceptional trajectory of the Greenland vs. zonal TCO in 2006 but no detailed analysis is presented to support that link, at least not until Section 6. Additionally, it looks to me like the increase is similar between the Greenland sector and the zonal mean also in 2013.

Response: We applied some clarification to respond to the first part as mentioned below. For the second part of the comment, that paragraph summarizes the increase rate of ozone during the SSWs. And 2006 is the only case that exhibits lower ozone increase over Greenland compared to the zonal average, and that's why we provided more explanation on that year. 2013, on the other hand, is consistent with other years as the TCO increase is higher over the Greenland sector. During 2013, EPV is higher than the climatology and ozone is dramatically lower than the climatology (-40 to -20 days of the event) over the Greenland sector.

The text is updated:

"The relative increase in TCO over the Greenland sector (blue line) is higher compared to the zonal average (orange line). The Greenland sector is climatologically inside the polar vortex area and has a lower TCO value during strong polar vortex which consequently exhibits higher relative increase after the vortex break down and mixing. However, dynamically disturbed winters such as years with minor SSWs before the major SSWs hinder the higher relative TCO increase over the Greenland sector compared to the zonal average. For instance, in 2006, the polar vortex weakened around 25

days before the major SSW (first column Figure 7, TCO 2006) due to a minor SSW, which coincides with the averaged TCO (solid line) increase compared to the climatology (dashed line) as seen in the second column Figure 7 (TCO 2006). The earlier timing of the positive anomaly caused a lower value in the TCO change after the event. The relative TCO increase over the Greenland sector exhibits a higher value during elongated polar vortex SSWs with 37% in 2018 and 31% in 2009. More details of physical mechanisms that cause variability in ozone during SSWs is discussed in section 6."

**Spc31:** P18 L15. Is the zonal average taken between 60N and 80N as before? Are these area-weighted averages? Are the anomalies calculated with respect to the climatology?

Response: Yes, the average is area-weighted and are based on climatology of non-SSWs years.

The text is updated:

"Figure 8 shows the temporal evolution of the vertical structure of ozone as a cross-section of area-weighted ozone anomalies for both the zonal average (60ºN to 80ºN) and the Greenland sector from 40 days before to 60 days after each SSW. The anomalies are estimated with respect to the climatology of non-SSW years between 2004 to 2019."

**Spc32:** P18 L25-27. "...because of". Again, causality is stated but not demonstrated until later.

Comment: The text is updated:

"The impact on ozone with the shortest duration occurred in 2008, which has multiple disturbances in the circulation and the shortest duration of easterlies (Table II)."

**Spc33:** P19 L15. I don't think that it's the vertical advection that leads to the poleward transport of PV. Isn't it rather due to planetary wave breaking and resulting mixing of low-PV low latitude air into the high latitudes?

Response: Thank you for noticing this. It is now clarified in the text

"The cyclonic polar vortex during wintertime is generated in response to the seasonality of radiative cooling. The intensified wave forcing before the SSW is manifested by both accelerated tropical upwelling and polar downwelling, and by poleward *eddy* transport of low EPV air parcels."

**Spc34:** P20 L20 The notation $M_y/dy$ is incorrect. If anything it should be $\partial M_{(")}/\partial y$. Same for the vertical derivative. More troubling is the use of "y" (which was never defined, I believe; the equations so far were in terms of $\varphi$). These two quantities should be the terms of the del operator acting on **M** in the spherical coordinates, i.e. one would expect something like

$$\frac{1}{a\cos\varphi}\frac{\partial}{\partial\varphi}\left(M_{(y)}\cos\varphi\right).$$

. Please make sure that the calculation is done correctly and that the notation is also correct.

Response: the text is updated to have the correct syntax of derivation: $e^{(z/H)}((a\cos\varphi)^{-1}\partial(\cos\varphi M_{(y)}/\partial y)$. Please note that My and Mz are already defined in eqs (4)-(5).

**Spc35 :** P21 L8. Is this connection with "elongated vortex" something that you could explain, or at least provide a viable hypothesis for?

Response: We show that vertical advection is more magnified during elongated vortex cases, which could be connected to stronger wave forcing in these cases. Occurrences of more elongated vortex SSWs in future could shed light on this matter and help us to make more robust conclusions.

In section 6, page 23:

> "This study shows that the larger geographical extent and magnitude of ozone changes during SSWs with elongated polar vortex is tied to greater vertical advection during these events."

Sec 7, page 27

> "The intensified vertical advection during elongated vortex events and abrupt wave forcing in these events is tied to the more intense magnitude and larger geographical extent of ozone changes during these events."

> The intensified vertical advection and abrupt wave forcing in during elongated vortex events is tied to the more intense magnitude and larger geographical extent of ozone changes during these events

**Spc36:** P22 L5. "*errors in the vertical derivatives over high latitude can be large as cos(φ) gets small)*" Why? I would think that it's the horizontal derivative that would be sensitive to that, but it's very likely that I'm missing something obvious.

Response: Yes, horizontal derivatives such equation in Spc34 is in mind. The text is updated.

**Spc37:** P22 L24. Again, it's not clear to me how significance is calculated.

Response: Significant bias is defined if the bias value is higher than the magnitude of the standard deviation. The text is edited:

> "a non-significant negative mean bias (bias is less than the standard deviation) exists at all sites (-8% to 15%)."

**Spc38:** P24 L23-25. As this (true) statement is not substantiated by the results of this study, it would be good to cite something here to support it.

Response: (Randel et al., 2002) is added to reference and support the statement.

**Spc39:** P25 L8. Even adding just the 2021 SSW would help!

We believe that by using the suggestions and feedback by both reviewers we have clarified the novelty of the work using six persistent events between 2004 to 2020. We briefly looked into the SSW in 2021 and considering its duration of 15 days of easterly winds with winds only reaching -10 m/s, it was a weak event compared to our original set of events. Thus, we decided to keep the scope of this study between 2004 to 2020.

**Technical corrections**

TC1: P3. L1 Please double check the grammar. "SSWs" is plural

Response: Applied.

> "Sudden stratospheric warming events (SSWs) are the largest alterations of stratospheric circulation during wintertime and the significant factor in the interannual variability of stratospheric transport."

Tc2: P3. L18-19. Grammar. "one of the strongest"à"some of the strongest"

Response: updated.

Tc3: P10 L5. The "a" (earth's radius) should be italicized.

Response: Updated.

Tc4: P10 L10-11. The equation numbering changed from 1, 2,.. to 2.4, 2.5. Please, fix it.

Response: Updated.

Tc5: P9 L22. "in this case ozone mixing ratio **tendency**"

Response: Updated.

Kris Wargan

---

## Author Comment (AC2)

Authors response:

We are thankful to both reviewers for their thorough comments. We have added a few paragraphs to the introduction to better clarify previous studies that are relevant to this manuscript and to elucidate remaining research questions that this study focuses on. A few graphs have been updated to have higher resolution and to include statistics, as suggested by the reviewers. We have used more clear language to highlight the novelty and main findings of this study. We applied most of the reviewer's suggestions and provide justification for those that we did not include. We believe that the quality of this manuscript has been greatly improved by these reviews.

Our detailed responses are given below, as well as comments on the full manuscript.

To facilitate reading of our responses and tracking changes in the full manuscript, we have labeled each of the reviewer's comments using the following labels:

List of labels:

reviewer 1 (Kris Wargan) :
    General comments (GC+#)
    Specific comments (Spc+#)
    Technical corrections (Tc+#)
reviewer 2 (Simon Chabrillat):
    General comments (GCC+#)
    Tables and figures comments (TF+#)
    Minor Comments and typos (MC+#)

We also used different fonts, sizes and colors: Reviewer comments are in black in Calibri font, our written responses are in blue in Calibri font, and the updated text in the manuscript is in cyan in Time New Roman font using a small font size.
* * *
Review of

Analyzing ozone variations and uncertainties at high latitudes during Sudden Stratospheric Warming events using MERRA-2

submitted to ACP by Bahramvash Shams et al. (doi: 10.5194/acp-2021-646, 2021)
S. Chabrillat, BIRA-IASB, October 2021

General Comments

This study provides an interesting overview of the 6 SSW events which happened over the Arctic since 2004. Its main strength is the consistent usage of a leading reanalysis (MERRA-2) to compare the dynamical variabilities of these events and their relationships with the corresponding distributions of ozone. All analyzed fields (temperature, winds, ozone) come from the same reanalysis system, for all 6 events as well as for the climatology extracted from years with no SSW. This provides good confidence in the methodological consistency and in the validity of comparisons across the different years. This contribution to the field is sufficiently substantial to warrant publication in ACP after some revisions as outlined below.

These revisions may be considered minor because they probably do not require any new calculation (yet, see major comments 2 and 4). A general revision of the text is certainly necessary to address the first major comment.

**Major Comments**

The text should be improved w.r.t. consideration of related work and appropriate references. Citations do not seem very well used: it is difficult to see the links between specific results and specific references because these are always provided in groups. Not being an expert on SSWs, I often wondered what results are new and what results have already been published (e.g. for specific years or using less consistent datasets).

Here are a few ideas and missing references to remedy this shortcoming:

GCC1: How does MERRA-2 relate with other available reanalyses? ACP has a whole special issue about the SPARC Reanalysis Intercomparison Project (S-RIP) explaining that several similar reanalyses are available (Fujiwara et al., ACP, 2017) with different performances w.r.t. ozone (Davis et al., ACP, 2017) and providing, on a topic related to SSW, an assessment of ozone mini-hole representation in reanalyses over the Northern Hemisphere (Millán and Manney, 2017). The CAMS reanalysis (Inness et al., ACP, 2019) assimilated very similar ozone data as MERRA-2 and has been extensively validated (Wagner et al., doi:10.1525/elementa.2020.00171, 2021).

Response: We added a sentence to show results from Davis et al., ACP, 2017, emphasizing that in the mid-stratosphere MERRA-2 has the best agreement with observations compared to other reanalysis data. However, their comparison study does not focus on northern high latitudes, which emphasizes the value of the comparison portion of our study. In this paper, we provide evidence to show the high accuracy of MERRA-2 stratospheric ozone estimation in the Arctic, which justifies our results and analysis.

"MERRA-2 is shown to have the best agreement with stratospheric ozone observations compared to other reanalysis data (Davis et al, 2017)."

Wagner et al. evaluated the CAMS reanalysis using some observations and concluded higher uncertainty in stratospheric ozone over high latitudes. Although the intercomparison of different reanalysis data is an important topic, we believe it is out of scope of this publication. A further study should include CAMS compared with other reanalysis data, similar to Davis et al., ACP, 2017.

GCC2: An extensive review about SSWs was published 8 months before the submission of this manuscript (Baldwin et al., 2021). Yet it is cited only to give the general definition of these events. This is a pity, because if would have be easy to contrast original results with results that are already discussed in this review.

Response: We agree with the general point of the reviewer with regard to highlighting the novelty and the need to clarify our research compared to previous studies. We added some paragraphs in the introduction that review previous work related to SSW and their impact on trace gases and then clarified the exact research questions that this study will focus on, including its novelties and how this study complements previous studies. However, the major discussion of Baldwin et al, 2021 concerns different theories and possible mechanisms that drive SSWs, as well as their impacts on different atmospheric layers. The discussion on stratospheric ozone is short and is only one small section of their paper. Thus, we have included citations of this paper with regard to the general definition of SSWs.

[revised manuscript text omitted]

GCC3: The manuscript nicely highlights the differences between elongated and displaced polar vortices prior to SSWs in the Northern Hemisphere. Has this distinction already been discussed w.r.t. ozone distribution in the Arctic stratosphere?

Response: We highlighted the novelty of the study more clearly throughout the manuscript. One of the novelties of this study is showing ozone changes due to SSWs are connected to the averaged polar vortex shape before the SSW, which led us to categorize them as either elongated or displaced. These averaged polar vortex indexes are different, though often related to, from split and displaced vortex breakdown conditions. As observed in the SSWs in 2018 and 2019, the polar vortex splits and the 15-day average polar vortex before that event is elongated. Other events such as those in 2013 and 2019, first displace and then split. However here we consider the displacement as the 15-day average, which is displaced and not elongated.

In our abstract:

"This study shows that the average shape of the Arctic polar vortex before SSWs influence the geographical extent, timing, and magnitude of ozone changes. The SSWs exhibit a more significant impact on ozone over high northern latitudes when the average polar vortex is mostly elongated as seen in 2009 and 2018 compared to the events in which the polar vortex is displaced towards Europe. Strong correlation ($R^2$=90%) is observed between the magnitude of change in average equivalent potential vorticity before and after SSWs and the associated averaged total column ozone changes over high latitudes. This paper investigates the different terms of tracer continuity using MERRA-2 parameters, which emphasizes the key role of vertical advection on mid-stratospheric ozone during the SSWs and the magnified vertical advection in elongated vortex shape as seen in 2009 and 2018."

"The averaged polar vortex state we refer to in this study is different, though often related to, split and displaced vortex morphology discussed in previous literature (e.g., Charlton and Polvani 2007). As seen during the SSWs in 2018 and 2009, in which the polar vortex split, the 15-day average polar vortex before those events is elongated. Other events, such as those in 2013 and 2019, first displace and then split. However, here we consider them displaced SSWs if the 15-day average EPV prior to the event is displaced and not elongated. Previous studies focused on the connection of the type of polar vortex breakdown to its impact on the speed of trace gas transitions (Charlton & Polvani (2007); Manney et al. 2009b). This study investigates the modulation of the magnitude and extent of ozone changes, and the results show that the average EPV shape before the vortex breakdown is more influential than the final form of polar vortex breakdown."

This study shows that the larger geographical extent and magnitude of ozone changes during SSWs with elongated polar vortex is tied to magnified vertical advection during these events.

"The variability in impact of SSWs on high latitude ozone is analyzed, two different patterns are found, and the possible dynamical mechanisms involved, are studied."

"This study shows that the averaged vortex shape before the SSWs is an important modulator of the magnitude and extent of ozone changes over high northern latitudes."

GCC4: The manuscript also highlights "the key role of vertical advection on mid-stratospheric ozone during the SSWs". Doesn't vertical advection also play a key role on mid-stratospheric ozone at other times and in other regions? What references have discussed this question?

Response: The global distribution of ozone concentration in the stratosphere cannot be understood without the influence of the Brewer-Dobson circulation (e.g., Hartmann and Garcia, 1979; Garcia and Solomon, 1983; Plumb, 2002), the advective part of which can be described by the residual velocities ($v^*,w^*$). In fact, its existence was first hypothesized by Brewer (1949) and Dobson (1956) trying to understand the distribution of ozone and other trace gases in the stratosphere.

The point of our paper is that during SSWs, the rapid changes in ozone are dominated by alterations in the vertical advection and horizontal eddy transport ($M_y$) rather than by other possible influences (horizontal advection, vertical mixing, chemical reactions, etc.)

Brewer A. W. (1949): Evidence of a world circulation provided by the measurements of helium and water vapor distribution in the stratosphere- Q. J. R. Meteorol. Soc., 75, 351-363.

Dobson, G. M. B. (1956): Origin and distribution of the polyatomic molecules in the atmosphere. Proc. R. Soc. London, Ser. A 236, 187-193.

Garcia, R. R. and Solomon, S.: A numerical model of the zonally averaged dynamical and chemical structure of the middle atmosphere, J. Geophys. Res., 88, 1379

Hartmann, D. L. and Garcia, R. R.: A Mechanistic Model of Ozone Transport by Planetary Waves in the Stratosphere, J. Atmos. Sci., 36, 350–364

Plumb, R. A.: Stratospheric Transport, J. Meteorol. Soc. Japan. Ser. II, 80, 793–809, https://doi.org/10.2151/jmsj.80.793, 2002.

GCC5: P.5, line 21 that "...temperature, the northward wind (v), vertical pressure velocity ($\omega$), potential temperature ($\theta$, calculated from temperature and pressure), and potential vorticity (PV) are extracted from the pressure-level MERRA-2 dataset."
The pressure-level dataset has a coarser vertical resolution than the model-level (i.e. sigma-pressure) dataset. Since the TEM analysis (Figs 11-12) involves vertical derivatives, it should be performed on dynamical variables which are retrieved on model levels. Is it the case?

Response: To have the finest vertical resolution for the comparisons with observations, MERRA-2 ozone at the model levels is used (GMAO, 2015a). To investigate dynamical mechanisms, using pressure-level data facilitates estimation of variables such as PV and potential temperature and consequently ozone at the pressure levels is used for associated analysis. We converted the pressure coordinate to geopotential height to estimate the derivatives for further analysis.

"To have the finest possible vertical resolution for the comparisons with observations, MERRA-2 ozone at the model levels is used (GMAO, 2015a). Other dynamical variables such as temperature, and the northward and vertical wind velocities (v, $\omega$), are extracted from the pressure-level MERRA-2 dataset (GMAO, 2015b), which facilitates the calculation of variables such as potential vorticity (PV) and potential temperature ($\theta$)."

GCC6: Figures 3 and 4, and related discussion (especially p.13, lines 12-17): How different are the corresponding diagnostics for years with no SSWs? Maybe one would obtain exactly the same biases and standard deviations of differences?
On these figures and throughout the text: the usual terminology is not "difference ratios" but "relative differences" or "normalized mean biases" and "standard deviations of the differences". See e.g. Lefever et al. (2015, doi:10.5194/acp-15-2269-2015).

Response: We agree that using relative differences is best and have updated the terminology to relative differences throughout the whole text.

For the comparisons, we had performed the comparison of non-SSW years in our preliminary stage of this study and the standard deviation of stratospheric ozone are lower and over the tropospheric layer the same level of uncertainties is seen. However, as during SSWs ozone fluctuations are larger, this study focusses on comparison of MERRA-2 and observation during these altered situations. This specific validation justifies our analysis of using MERRA-2 for analyzing ozone during highly

variable periods and on the other hand the high accuracy of data during the highly fluctuated time also prove the performance of non SSW years.

GCC7: P.19, lines 4-5: "The positive temperature anomalies in mid stratospheric layers start a few weeks before the SSWs in the 4 cases of a displaced vortex (2006, 2008, 2013, and 2019)." But from the definition of the SSW (p.3 line 1) one of the two conditions to identify a SSW is an abrupt and intense increase of stratospheric temperature. Yet Figure 9 shows that the increase in temperature was not abrupt on these 4 years (those with displaced polar vortices). So one wonders how your algorithm for SSW identification could identify 2006/01/21, 2008/02/22, 2013/01/06 and 2019/01/02 ? I am also confused by the next sentence:
"On the other hand, the intrusion of the positive temperature anomalies to mid stratospheric layers is almost coincident with SSWs in the 2 elongated vortex cases."
But seeing the definition of SSWs, shouldn't this be a feature of all SSWs?
I think that this should be clarified not only in the author's response but also in the revised manuscript.

Response:

We clarified the definition of SSW as below:
> SSWs are defined by a reversal of the climatological westerly wind circulation, which typically coincides with an abrupt and intense stratospheric temperature increase"

Moreover, it should be noted that the definition is based on reversal of 60 N, 10 hpa zonal mean zonal wind. On the other hand, the cross-section plots are the anomalies of average temperature profile based on averages over the 60 to 80 N with respect to the climatology profile of non-SSW years. A substantial temperature increase (around 10-30 K) below 50 km is evident within approximately 10 days prior to all events in the zonal average.

**Tables and figures TF1**

TF1: Table 1: What is the "full PCO" in "%full PCO uncertainties"? Does it mean "Partial Column of Ozone"? But for what pressure range? Or maybe that is the TCO?
Do these uncertainties (two rightmost columns) come from Bognar et al. (2019) or are they a new result of this paper?

Response: we defined the PCO in the main body text as partial column ozone. But to facilitate the read and comprehension of the table we also included it table caption. As comparison are limited by ozonesondes maximum height which is around 30 km on average.
The text is updated:
> full PCO is replace by "% PCO uncertainties (Ground -30 km)"

TF2: Figure 1: Add a box for the Greenland sector; stretch the color scale towards the reds in order to increase the contrasts

Response: Greenland box is added. For this plot we have used standard "viridis" color map. "viridis" is among linear space colormaps that prevent misinformation by using non linear space among colors and adding non realistic contracts. Moreover the hues used in "viridis" have unique pairs in grayscale which makes it readable for different color blindness.

TF3: Figure 2: Add a sentence to the caption, e.g. "**The vertical red lines highlight the dates of the 2008 and 2009 SSWs**".

Response: we added the quat to the caption.

TF4: Figures 3 and 4: re-formulate the captions to obtain similar captions while avoiding the words "difference ratios". Consider: "**Normalized mean biases and standard deviations of MERRA-2 with respect to ozonesondes/FTIR observations** ".

Response: we updated the text throughout the whole text and captions to use relative difference. And the caption of figure 4 use previous caption:

> Same as Figure 3 but for relative differences of FTIR retrieved ozone from MERRA-2. Statistical summaries ofthe MERRA-2 and NDACC comparisons in four layers of ground to 8 km, 8km-15km, 15km -22km, and 22km- 30kmfor each station are shown on top of each plot.

TF5: Figure 5: totally unreadable, even at maximum zoom on a large screen! You must change the layout of the figure to decrease the margins around each map and increase its relative area and increase the resolution of the bitmap or (better) save these maps as vector-oriented graphics (PDF). Please clarify the caption: "Mean values of the TCO anomalies..."

Response: the figure is updated and has a higher resolution! The caption is clarified.

> The anomaly TCO average over 15 days prior (alphabet1, first and third columns) and 15 days after to each SSW (alphabet2, second and forth columns) compared to climatology on non SSW years.

TF6: Figure 8: Please remind in the caption, for the casual reader: "...averaged over  **latitude band 60°N-80°N** and **over the** Greenland sector **(60°N-80°N, 10°W-70°W)**."

Response: it is applied.

TF7: Figure 10: How different is the time evolution of w* on a year with no SSW? One expects smaller and less perturbed values, but by how much? Consider adding the same figure but from the climatology of years with no SSW

Response: In the NH, the vortex is very rarely undisturbed, for the wave forcing is relatively strong (at least compared to the SH) throughout the winter. However, the values of w* 40 to 30 days before SSWs in Fig. 10 should be characteristic of an "undisturbed" vortex situation.

TF8:  Figure 11: these plots should not show results below 15 km because these results cannot be discussed since

"Considering the larger uncertainties of ozone estimation in MERRA-2 below 15 km, and the possibility of larger uncertainties in dynamic parameter estimations, this study does not analyze the impact of the dynamics on ozone in the lower stratosphere." (p. 21, lines 17-20).

Response: The plot is updated to 15km to 30 km.

**Minor Comments and typos MC1**

MC1:  P.1, line 17: clarify "**During SSWs**, changes in..."

Response: the text is updated.

MC2:  P.3, lines 10-11: "... many other factors such as lower stratosphere conditions, the geometry of the polar vortex, the gradient of potential vorticity (PV) at the edge of the polar vortex, and synoptic systems at lower altitudes (Tripathi et al. 2015, de la Cámara et al., 2019; Lawrence and Manney, 2020). Changes in momentum deposition associated with these processes leads to..." These are not "processes". Maybe "conditions" or "dynamical states"?

Response: the text is updated.

"...dynamical states lead to ..."

MC3:  P.4, lines 15-16: improve transition with next paragraph, e.g.
"This study investigates **dynamical variability and** ozone variations **above the Arctic** (between 60°N and 80°N) **both** in the zonal average **and above a specific region**, during six SSWs using the MERRA-2 dataset."

Response; the whole paragraph is revised as discussed in response to GC3.

MC4:  P.4, line 20: this is the first occurence of "Greenland sector" so you should move here your definition of this region (currently on p.16) and also draw the corresponding box on Fig. 1.

Response: it is added.

**MC5:** P.5, lines 7-10 and 22-23: this attempt to define the MERRA-2 reanalysis and its assimilation system is not correct. Reanalysis systems use only one model, here GEOS5; a well-designed reanalysis does not have any variations in models nor in methods of analysis – only in assimilated datasets of observations. See e.g. Fujiwara et al. (ACP, 2017) for a general yet correct description of reanalysis systems such as MERRA-2.

Response: the text is updated to apply the reviewer comment:

"A variety of data sets are incorporated into a general circulation model to create 3-dimensional MERRA-2 ozone datasets with a time-frequency of 3 hours (Wargan et al., 2017; Gelaro et al., 2017)."

"In reanalysis products such as MERRA-2, methods of analysis, model uncertainties, and observations cause uncertainties in the products (Rienecker et al. 2011)."

**MC6:** P.6, lines 5-6:
"...atmospheric dynamics, displaced/split polar vortex, and hemispherically asymmetric conditions during SSWs may cause unusual nonlinearity in ozone flux/transport terms." What do you mean by "unusual nonlinearity"?

Response: We clarified the text.

"Moreover, the anomalous atmospheric dynamics, displaced/split polar vortex, and hemispherically asymmetric conditions during SSWs may cause complexity and additional uncertainties in estimation of ozone flux/transport terms."

**MC7:** P.7, lines 16-18: "Having the ability to resolve the fine structure of solar radiation spectra allows the retrieval of a variety of trace gases using the NDACC solar FTIR. However..."
This sentence is irrelevant for this paper – consider removal.

Response: As suggested we removed the sentence.

**MC8:** P.7, line 21-22: this citation of Bognar et al. (2019) does not seem to belong here as they validate satellite instruments – not ground-based instruments?

Response: As suggested we removed Bognar et al. (2019).

**MC9:** P.8, line 1: "More details  **on** the ozone retrievals at Eureka"

Response: the sentence is updated.

**MC10:** P.8, lines 16-17: "However, the vertical resolution of the remote sensing retrieval is often not similar to the model grid points...". Consider instead:

"Since the the vertical resolution of the remote sensing retrieval is **much coarser than the vertical resolution of the model**..."

Response: This is a general statement, as some remote sensing data such as MLS have high vertical resolution, however it might not match the model levels, for one-to-one comparison. We rephrased the sentence as below:

> Since, the vertical resolution of the FTIR retrieval does not match to the vertical resolution of the assimilation system.

MC11:  P.8, eq. (1): Do $x_s$, $x_h$ and $x_a$ represent vertical profiles of ozone **mixing ratios**? Please clarify.

Response: we updated the text as bellow: where $x_s$ is the final smoothed profile, $x_h$ is the reanalysis estimated profile, and $x_a$ and A are the a priori and averaging kernel of ozone mixing ratio for the retrieval respectively.

MC12:  P.8, line 26: "The smoothing method effectively linearizes the ozone from the model..."

I do not understand. How can ozone, or even its mixing ratio, be "linearized" ?

Response: we rephrased the sentence to clarify.

> The smoothing method effectively applies the sensitivity of the retrieval to the ozone mixing ratio profile from the reanalysis using the averaging kernel and the priori information to create comparable profiles. (Rodgers and Connor, 2003).

MC13:  P.10, lines 15-16: "the lack of net chemical production in the assimilation model should not dramatically impact our conclusions."
Could heterogeneous chemistry losses happen in Polar Stratospheric Clouds prior to the SSWs?

Response: The chlorine activation requires sunlight, so during Nov, Dec, and Jan when dominant parts of the high latitude is dark, these reactions are small. The significant ozone depletion in the Arctic occurs in undisturbed springtime when both conditions of cold stratosphere and sunlight are met. The text is updated to clarify more.

> "The contribution of dynamical and chemical drivers of ozone anomalies varies throughout the year. During springtime, both dynamical resupply and chemical depletion strongly modulate ozone changes. Assuming an isolated polar vortex and neglecting isentropic mixing, a previous study showed a similar magnitude of influence from chemical ozone depletion processes and dynamical ozone supply during the springtime (Tegtmeier et al. 2008). However, Strahan et al. (2016) used a chemistry and transport model to show that dynamical processing affects ozone changes by a factor of two more than chemical processing during March. However, chemical processes are not significant drivers of ozone changes in the middle stratosphere from November to February in the Arctic because of the polar night (de la Cámara et al. 2018b). Moreover, it has been shown that during years with SSWs, Arctic ozone depletion is significantly diminished (Strahan et al. 2016). However, if prior to or during the SSWs, the polar vortex moves outside of the region of the polar night (to lower latitudes), ozone depletion will occur as shown in the 2013 SSW by Manney et al. (2015). By limiting our analysis to latitudes between 60ºN to 80ºN, this impact is minimized in our analysis."

MC14: P.11, lines 15-20: this is a pure repetition of details already given in the caption of the figure. I recommend to keep this in the figure caption and to remove it here.

Response: we removed the repetitive text and here is the updated sentence:

The results and statistics of comparisons between ozonesondes and MERRA-2 are depicted as the relative differences in Figure 3.

MC15: P.11, lines 23-24, consider:
"the 5km-10km **layer** includes **the** upper troposphere **and the** lower**most** stratosphere (UTLS), while the 10-30km layer includes the lower **middle** stratosphere."

Response: It is applied.

MC16: P.12, line 2: "MERRA-2  **appears** unable to retrieve..."

Response: based on other suggestions, this sentence is rephrased as:

The extreme low ozone values near the surface are not represented in MERRA-2 as it does not include bromine chemistry.

MC17: P.13, line 1: "...contain the most column ozone...".

Consider instead: "...**contribute most to the total ozone column**..."

Response: It is applied.

MC18: P.13, line 24-25: re-write the sentence. Consider e.g. :
"...our primary analysis is focused on the mid-stratospheric layers **which contribute most to the TCO and**  where the measurements are most reliable "

Response: It is applied.

MC19: P.14, lines 8-12: please split this sentence in several parts to make it clearer. Specifically, I understood only later on that you plot and discuss **the mean values over the 15 days prior to the SSW and over the 15 days that follow**. Please state this clearly already here and also in the caption of Fig.5.

Response: It is applied.

MC20: P.14, lines 22-23: do you mean
"The easterly winds lasted 16 days  **after** the major warming on 22 February." ?

Response: It is applied.

MC21: P.16, line 8: what do you mean by "semi-symmetrical shape"?

Response: it is updated to a Fairly symmetrical shape around the arctic.

MC22: P.16, lines 21-23: if we zoom on Fig.15 to the maximum possible and on a large screen, we can see (despite the insufficient resolution of the figure) that this characterization does not hold for the 2019 SSW. See also major comment on unreadability of Fig. 5.

Response: The figure is updated to have higher resolution. The text is updated to emphasize on the importance of Greenland sector as climatologically being isolated by polar vortex to provide an image of regional impact of SSWs in comparison to the zonal average.

> As the Greenland sector is one of the critical regions that is climatologically isolated by the polar vortex

MC23: P.17, line 3: "the Greenland sector exhibits a very strong isolated stratospheric air circulation during wintertime..." Consider instead: "**The air masses above the Greenland sector are more strongly isolated than at other Arctic longitudes**".

Response: It is updated

MC24: P.17, line 7: "...and the Greenland sector  from 40 days before to 60 days after each SSW..."

Response: It is updated

MC25: P.17, line 14: "The climatological polar vortex position is located over the Greenland Sector (Figure 1)". I do not agree: the center of the climatological vortex, as shown on Figure 1, is above Ny Alesund which is slightly outside (East) of the Greenland sector as defined here.

Response: we updated the text to be more clear. It is True that Ny Alesund is at the center of the climatology of the polar vortex, and Greenland sector is one area that is located inside the polar vortex. The location of Ny Alesund is very close to the area that we used for the Greenland sector thus in the mid stratospheric analysis it follows the same pattern for the most part.

> The Greenland Sector is located inside the climatological polar vortex area is located over (Figure 1)

MC26: P.17, lines 18-19: "... over the Greenland sector  **with a** larger drop  of EPV **in this region than in**  the zonal **mean**."

Response: It is updated.

MC27:  P.18, lines 1-2: "In all SSWs  **relative increase** of the TCO is higher over the Greenland sector compared to the zonal average with the exception of 2006..." Is this significant? It seems to me that the values above Greenland are barely larger than at other longitudes.

Response: the wording is updated. Commenting on the significance of the relative difference over Greenland is difficult to make. The Greenland sector as an area under the climatological polar vortex has lower TCO during the undisturbed winter with strong polar vortex. Thus, after the polar vortex break down and transition of enriched ozone air to high latitude it exhibits a higher relative TCO increase. However, in some study cases, during years with minor SSWs prior to the major SSWs, the TCO fluctuates even over the climatological polar vortex area which prevents a significant difference in relative increase of TCO over the Greenland compared to zonal average.

Our conclusion is on the significant influence of elongated cases on relative TCO increase over the Greenland sector compared to displaced cases.

"Compared to the 40-day average of TCO prior to the SSW, the highest percent zonal TCO increase of 29% is observed for one of the elongated polar vortex SSWs in 2009. The relative increase in TCO over the Greenland sector (blue line) is higher compared to the zonal average (orange line). The Greenland sector is climatologically inside the polar vortex area and has a lower TCO value during strong polar vortex which consequently exhibits higher relative increase after the vortex break down and mixing. However, dynamically disturbed winters such as years with minor SSWs before the major SSWs hinder the higher relative TCO increase over the Greenland sector compared to the zonal average. For instance, in 2006, the polar vortex weakened around 25 days before the major SSW (first column Figure 7, TCO 2006) due to a minor SSW, which coincides with the averaged TCO (solid line) increase compared to the climatology (dashed line) as seen in the second column Figure 7 (TCO 2006). The earlier timing of the positive anomaly caused a lower value in the TCO change after the event. The relative TCO increase over the Greenland sector exhibits a higher value during elongated polar vortex SSWs with 37% in 2018 and 31% in 2009. More details of physical mechanisms that cause variability in ozone during SSWs is discussed in section 6."

MC28:  P.18, lines 5-12: is this paragraph useful or interesting? Consider deletion.

Response: It is deleted.

MC29:  P.18, lines 22-23: split the sentence: "...for both the zonal averaged and the Greenland sector. As expected the structures of ozone anomalies are smoother in the zonal average compared to the Greenland sector ."

Response: It is updated.

MC30:  P.18, lines 24-25: this sentence is very unclear. Are you still comparing the Greenland sector with the zonal average? Please re-write.

Response: As we removed a paragraph that talked about recovery based on your suggestion (MC28:). This sentence is also irrelevant and is removed.

MC31: P.18, line 26: "The shortest impact on TCO  **happened in** 2008..."

Response: It is updated.

MC32: P.19, lines 15-17: "...which leads to **poleward** advection of low EPV air parcels poleward. The conservation of EPV causes anticyclonic circulation, which gradually drives easterly the zonal mean zonal winds, and leads to **the** displacement or splitting of the polar vortex."

Response: It is updated.

MC33: P.20, line 1, consider: "Occurrences of minor SSWs  **can be seen through** the early appearance"

Response: It is updated.

MC34: P.20, line 8, consider: "The suppressed wave activity  **leads** to the recovery of..."

Response: It is updated.

MC35: P.22, line 7: Section 7 provides a summary, with only the last paragraph providing a conclusion. Hence the title of this section should be "**Summary and conclusion**".

Response: It is updated.

MC36: P.22, line 13: "**The** MERRA-2 reanalysis..."

Response: It is updated.

MC37: P.22, lines 24-25, consider: "From 5km to 10km,  negative mean bias exists in all sites (-8% to 15%) **but it is not significant due to the larger standard deviation.**"

Response: It is updated.

MC38: P.23, line 1: please describe "G-5km (<20%)" properly, with words

Response: It is updated.

Around 20% standard deviation of relative differences is observed at G-5 km.

**MC39:** P.23, line 8: "These results emphasize the high quality of MERRA-**2 at least after 2004, the year when MLS data became available**." This is important!

Response: It is updated.

**MC40:** P.23, lines 9-10: "Higher uncertainties in the UTLS are also expected because MLS has **a dominant contribution in the MERRA-2 reanalysis and a** lower sensitivity at lower altitudes "

Response: It is updated.

**MC41:** P.23, line 22: "The TCO increases rates and the magnitude of changes in EPV after these cases are large and the intrusion of positive temperature anomalies to the mid stratosphere is coincident with **these** SSWs dates."

Response: It is updated.

**MC42:** P.24, lines 3-5, consider: "A strong **cor**relation is observed between the magnitude of change in the averaged EPV  around the SSW, and the magnitude of TCO change for the same period for all six studied SSWs."

Response: It is updated.

**MC43:** P.24, lines 6-15: "The Greenland sector is one of the critical regions that is impacted by negative TCO **anomalies** before the elongated polar vortex in 2009 and 2018; positive TCO **anomalies** occurs before displaced SSWs. To identify the similarities and differences of zonal versus the regional impact of SSWs on ozone, the analyses are applied over the Greenland sector as well as the zonal average. The general structure of the vertical ozone anomaly **over** the Greenland sector is similar to the zonal **structure**. However, as expected the ozone anomaly over the zonal average is smoother than over the Greenland sector which results in a more magnified TCO increase over Greenland. The increased rate over the Greenland sector is between 15% in 2006 to 38% in 2018, while the zonal average ranges between 8% in 2008 to 29% in 2009. Moreover, **the** TCO exhibits a faster recovery to the climatology values over this region compared to the zonal average."

Response: It is updated.

**MC44:** P.24, line 26: not understandable – please re-write:
"The faster recovery of zonal temperature and ozone at middle stratosphere within 30 days is recorded for 2008 with the shortest duration easterly zonal mean zonal winds."

Response: It is updated.

The fastest recovery of zonally averaged temperature and ozone at middle stratosphere happen in 30 days for 2008.

MC45:  P.25, line 3-6: "In conclusion, the MERRA-2 dataset is shown to capture **the** ozone variability in the middle stratosphere and provides dynamical information to investigate the impact of SSWs. The impact of SSWs on ozone and the role of vertical advection is shown to be more intense in 2009 and 2018 with an elongated polar vortex compared to **the** displaced  **vortices** in 2006, 2008, 2013, and 2018."

Response: It is updated.

MC46:  P.25, last sentence: "the dramatic ozone increases over high latitudes during SSWs points to consequences of these events on the global earth system and possible environmental/ecosystem changes that could be investigated in future studies."
This is a quite vague statement, and I am skeptical as the timescales for environmental/ecosystem changes are much longer than those due to SSW perturbations. Maybe it is possible to conclude instead with the changes in SSW occurences that are expected from climate change?

Response: we updated the final statement based on suggestion:

"Although there is no consensus across future climate simulations on whether SSW occurrences will increase or decrease in response to increased greenhouse gas concentration (Ayarzaguena et al. 2018, 2020), many simulations show a significant change. The dramatic ozone increases over high latitudes during SSWs points to the consequences and implications for ozone if the rate of SSW increases in future."